# Earth-Abundant Electrocatalysts in Proton Exchange Membrane Electrolyzers

**Xinwei Sun [1], Kaiqi Xu [1], Christian Fleischer [1], Xin Liu [1], Mathieu Grandcolas [2]** **,**
**Ragnar Strandbakke [1], Tor S. Bjørheim [1], Truls Norby [1] and Athanasios Chatzitakis [1,\***]

[1]  Centre for Materials Science and Nanotechnology, Department of Chemistry, University of Oslo, FERMiO, Gaustadalléen 21, NO-0349 Oslo, Norway; xinwei.sun@smn.uio.no (X.S.); kaiqi.xu@smn.uio.no (K.X.); christian.fleischer@smn.uio.no (C.F.); xin.liu@smn.uio.no (X.L.); ragnar.strandbakke@kjemi.uio.no (R.S.); t.s.bjorheim@kjemi.uio.no (T.S.B.); truls.norby@kjemi.uio.no (T.N.)

[2]  SINTEF Materials and Chemistry, P.O. Box 124 Blindern, NO-0314 Oslo, Norway; mathieu.grandcolas@sintef.no

\*   Correspondence: a.e.chatzitakis@smn.uio.no; Tel.: +47-228-40-693

**Abstract:** In order to adopt water electrolyzers as a main hydrogen production system, it is critical to develop inexpensive and earth-abundant catalysts. Currently, both half-reactions in water splitting depend heavily on noble metal catalysts. This review discusses the proton exchange membrane (PEM) water electrolysis (WE) and the progress in replacing the noble-metal catalysts with earth-abundant ones. The efforts within this field for the discovery of efficient and stable earth-abundant catalysts (EACs) have increased exponentially the last few years. The development of EACs for the oxygen evolution reaction (OER) in acidic media is particularly important, as the only stable and efficient catalysts until now are noble-metal oxides, such as $IrO_x$ and $RuO_x$. On the hydrogen evolution reaction (HER) side, there is significant progress on EACs under acidic conditions, but there are very few reports of these EACs employed in full PEM WE cells. These two main issues are reviewed, and we conclude with prospects for innovation in EACs for the OER in acidic environments, as well as with a critical assessment of the few full PEM WE cells assembled with EACs.

**Keywords:** polymer exchange membrane; electrocatalysts; noble metals; earth abundant elements; water splitting; acidic environment; oxygen evolution reaction; hydrogen evolution reaction; anode and cathode electrodes

## 1. Introduction

Currently, 81% of the global energy demand is met by fossil fuels and it is estimated that more than 540 EJ was supplied for the total global energy demand in 2014. This figure is expected to increase by 40% towards 2050 [1]. The $CO_2$ emissions from combustion of fossil fuels are large enough to severely alter the Earth's climate and global ecosystem, forcing mankind to accelerate the return to renewable energy. This is amplified by the reserves of fossil fuels estimated to last only 50–60 years [2–4].

Hydrogen ($H_2$) can meet our future energy demands as a clean and sustainable fuel, but cost-effective ways need to be developed for a successful turn towards the hydrogen economy [5–9]. Water electrolysis is an environment friendly scheme for conversion of renewable electricity (e.g., solar, wind) into high purity hydrogen, but at present electrolysis accounts for only 4% of the total hydrogen production [10]. The rest is covered by transformation of fossil fuels, such as natural gas steam reforming, coal gasification and partial oxidation of hydrocarbons [11–14], however, all these routes involve the release of $CO_2$. Polymer electrolyte membrane water electrolysis (PEM WE) has the advantages of simplicity, compact design, fast response, high current densities, production of ultrapure

hydrogen that can be electrochemically pressurized, and small footprint. The PEM WE concept was first investigated and demonstrated in the 1960s [15–17]. Since then, substantial research has been dedicated to improve the different PEM WE components, and as a result, this technology is approaching commercial markets [18]. What hinders the implementation of PEM WE on a large scale is its acidity, which necessitates the use of noble metals, such as Ir, Pt, or Ru as electrocatalysts. Additionally, acidic conditions are more preferable as the concentration of reactant protons is higher [19,20]. The high cost of the polymeric membrane is another obstacle. Currently, the capital investment cost (CAPEX) for a PEM WE system is around $1500 per kWe (kW electricity input) and the cost per kg of $H_2$ is $7.1, taking into account that the electricity is provided by renewables [21–23].

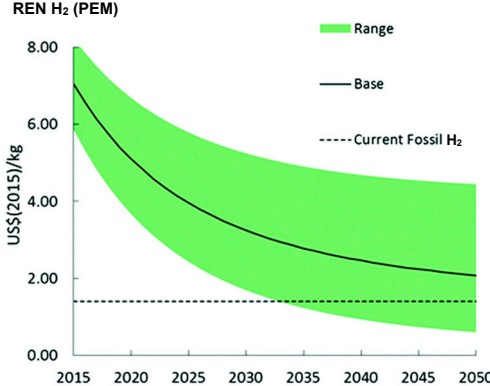

**Figure 1.** Learning curve for renewable PEM $H_2$ production showing the projected levelized costs until 2050 per kg $H_2$ in USD. Reprinted with permission from [24]. Copyright 2018, The Royal Society of Chemistry.

In comparison, the $H_2$ cost through steam methane reforming (SMR) is only $1.40 [25] and the optimistic break-even year for renewable PEM $H_2$ production based on learning curves is around 2033 (Figure 1) [24]. The same study underlines that the major cost of PEM lies in the electricity consumption [24]. This is directly connected to the overpotential required for efficient water electrolysis, i.e., the overpotential of the electrocatalysts to reach certain current densities. It is established that in terms of efficiency and stability, the platinum group metals (PGMs) are the best choices for electrodes in a PEM electrolyzers, however, the question is at what cost. For example, the annual global production of Pt in 2017 was $1.7 \times 10^5$ kg, while the total demand for Pt in the same year was over $2.2 \times 10^5$ kg. If the recycled Pt is also considered as part of the production, the annual production of Pt just met the total demand (USGS 2016 Mineral Years Report). Therefore, widespread installation of Pt-Ir based PEM electrolyzers will dramatically increase the total demand of PGMs. As an example, a Terra Watt (TW) hydrogen production system requires 0.5 and 10 years of annual global production of Pt and Ir, respectively [26]. One has also to take into consideration that Ir is typically produced as a minor by-product of Pt [27]. In other words, the annual production of Ir is also determined by the production rate of Pt. As a result, the increasing demands of Ir will increase their cost due to its dependence on Pt mining.

We performed our own calculations, using the state-of-the-art PEM electrolyzer that we will return to in Chapter 3. In this system, the cathode has 0.4 $mg_{Pt}/cm^2$ of Pt, and the anode 1.54 $mg_{Ir}/cm^2$ and 0.54 $mg_{Ru}/cm^2$ of Ir and Ru, respectively. Our calculations (see Supplementary Materials for more information) suggest that such a PEM system with a power density of 1.18 $W/cm^2$ requires 1.5, 180 and 12 years of annual production of Pt, Ir and Ru, respectively, to cover 1 TW of hydrogen production. It is evident, that the replacement of the noble metal electrocatalysts for both the hydrogen evolution reaction (HER) and oxygen evolution reaction (OER) will have a tremendous impact on the future scale-up activities for PEM WE. Furthermore, competition will be avoided with other industrial activities, such as the automobile and electronics sectors, where the demand for PGMs is big.

A wide range of earth abundant catalysts (EACs) for the HER in acidic, neutral and alkaline media have been developed and includes metal sulfides [28–33], metal phosphides [34–39], metal alloys [40,41], chalcogenides [42,43], as well as metal- and heteroatom-substituted carbon-based materials [44–46]. Some of these EACs show improved efficiencies and good endurance under strong acidic conditions [34,35,37,47,48] while others are not stable or they require large onset overpotentials [49–52]. The situation is even more challenging on the OER side, the bottleneck in overall water splitting, where the complex 4-electron process that produces protons and oxygen requires high overpotentials. Only noble-metal oxides such as $IrO_2$ and $RuO_2$ are efficient catalysts for the OER in acidic media, but the $RuO_2$ is unstable and deactivates rapidly [53,54], therefore the lack of cost-efficient alternatives to $IrO_2$ is the major challenge in the field of PEM-based water electrolysis.

This field of research is very active, and according to Web of Science, 2043 reports have been published during 2017 on both OER and HER catalysts (Figure 2). Motivated by these figures, as well as the challenging electrochemistry under the intense conditions required by the PEM WE, we wanted to see how many of these reports referring to EACs were actually applied in PEM WE devices, replacing in fact the noble-metal catalysts. Therefore, the main purpose of this article is not primarily an exhaustive report on EACs developed for the HER and OER in acidic conditions, which were tested and studied in half-cells, typically involving measurements in three electrodes with rotating disc electrodes (RDE), but to focus on those applied and tested in full PEM WE cells. Do the catalysts perform as expected from the half-cell measurements, or are there deviations related to differences in configuration, supply of reactants, deposition on porous substrates, leaching of electroactive elements (i.e., stability), and surface area exposed? Moreover, what are the recent advances on EACs for the OER under strongly acidic conditions? In the current article we document the very first reports on EACs for the OER in acidic environment, as well as one applied EACs-based PEM WE system.

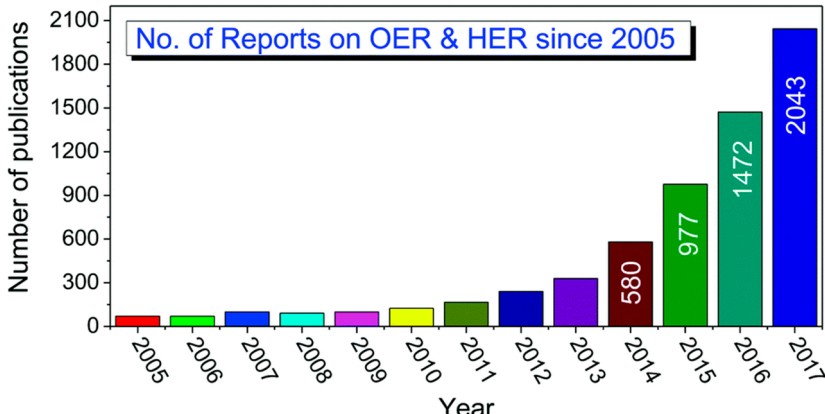

**Figure 2.** Histogram showing the number of scientific reports on OER and HER from 2005 to 2017. Reprinted with permission from [55]. Copyright 2018, The Royal Society of Chemistry.

## 2. Principles of PEM Water Electrolysis

The electrochemical conversion of water to hydrogen and oxygen is known as water electrolysis, and was discovered already in 1800 [56]. Since then, the use of two electrodes immersed in an aqueous caustic solution of KOH electrolyte, known as alkaline water electrolysis, was developed and utilized for industrial applications [57]. Although some improvements as current density and operating pressure are foreseeable [54], this well-established technology is still the most cost-effective choice for electrochemical hydrogen production on an industrial scale at present.

Another promising water electrolysis cell that operates at low temperatures (normally below 80 °C) is the proton exchange membrane (also known as polymer electrolyte membrane) (PEM) electrolyzers. The concept of PEM water electrolysis was idealized by Grubb in the early fifties [15,16] and first manufactured by the General Electric Co. in 1966 [17], where they take the advantage of a solid polymer perfluorinated sulfonic membrane as electrolyte for hydrogen production. Some typical

pros and cons for PEM water electrolyzers compared with the classic alkaline water electrolyzers are summarized in Table 1.

We highlight again that a cost reduction by developing earth-abundant electrocatalysts with comparable performance and a further improvement in the energy efficiency of the PEM water electrolyzers are essential factors before PEM WE becomes a competitive solution for large-scale hydrogen production.

**Table 1.** Advantages and disadvantages of PEM WE over alkaline water electrolysis.

| Advantages [17,54,58] | Disadvantages [58–60] |
|---|---|
| Compact system design | Acidic electrolyte |
| → Fast heat-up and cool-off time, short response time<br>→ Low gas cross-permeation. Withstands higher operating pressure across the membrane. Higher purity of hydrogen. Higher thermodynamic voltage<br>→ Easier hydrogen compression, facilitates hydrogen storage | → Higher manufacturing cost due to expensive materials and components, i.e., current collectors, bipolar plates, noble catalysts, membranes<br>→ Limited choices of stable earth-abundant electrocatalysts for the OER |
| Solid, thin electrolyte | Solid, thin electrolyte |
| → Shorter proton transport route, lower ohmic loss<br>→ Operates under wide range of power input | → Easily damaged by inappropriate operation (e.g., overheating) and cell design<br>→ Sensitive to imperfections, dust, impurities |
| Operates at higher current density | |
| → lower operational costs<br>→ Differential pressure across the electrolyte<br>→ Pressurizes hydrogen side alone, avoids danger related to pressurized oxygen | |

## 2.1. Operating Principles

When a PEM electrolysis cell is in operation, an excess of water is supplied to the anode, where water decomposes into protons, electrons and oxygen gas by the electrical energy (Equation (1)). The protons are transported to the cathode by passing through the polymer electrolyte, while the generated electrons travel along an external circuit and combine with the protons into hydrogen gas, as described in Equation (2). The amount of hydrogen gas generated is twice that of oxygen, as defined by the overall reaction, Equation (3), whereas $\Delta G^0$ is the standard Gibbs free energy of the net water splitting reaction.

Anode (OER)

$$H_2O \rightarrow 2H^+ + 2e^- + \frac{1}{2}O_2 \tag{1}$$

Cathode (HER)

$$2H^+ + 2e^- \rightarrow H_2 \tag{2}$$

Net water splitting reaction

$$H_2O \xrightarrow{\Delta G^0} H_2 + \frac{1}{2}O_2 \tag{3}$$

## 2.2. Thermodynamics

The standard theoretical open circuit voltage (OCV), also referred as standard reversible cell voltage, $U_{rev}^0$, required by water electrolysis under standard conditions, can be derived from; the standard Gibbs free energy ($\Delta G_R^0$) of + 237.2 kJ/mol $H_2$, Faraday's constant (F), and the number of electrons ($n = 2$) exchanged during water splitting under standard conditions; $p = 1$ bar, $T = 298.15$ K (Equation (4)) [61].

$$|U_{rev}^0| = |\frac{-\Delta G_R^0}{n \cdot F}| = 1.229 \; V \tag{4}$$

The positive Gibbs free energy change reflects that the water electrolysis reaction is thermodynamically unfavorable. For the reaction to proceed at finite rate, overpotentials for the OER and HER, as well as the electrolyte resistance, must be added to $U_{rev}^0$ [58]. These represent losses generating heat. At an overpotential of 0.25 V, i.e., an applied cell voltage of 1.48, V this heat balances the heat consumed by the reaction under standard conditions, and the cell operates in thermoneutral mode; 1.48 V is termed the thermoneutral voltage [62] and is reasonable to use when calculating the voltage efficiency of the cell. Thus, the actual operating cell voltage is the sum of all the different overpotentials (Equation (5)) [54,63].

$$U_{op} = U_{rev}^0 + \eta_a + \eta_c + \eta_{el} + \eta_{sys} \tag{5}$$

$U_{op}$ is the operational voltage, $U_{rev}^0$ is the standard reversible potential, $\eta_a$, $\eta_c$, $\eta_{el}$ and $\eta_{sys}$ are the overpotentials related to the anode, cathode, ionic conductivity of the electrolyte membrane, and system losses (resistance in contacts, interconnects, current collectors, wires, etc.), respectively. It should be emphasized that the half-reactions described in Equations (1) and (2) are simplifications of more complex multistep and parallel electrochemical reaction pathways [64].

### 2.3. Main Cell Components and Requirements

The core component of a PEM electrolysis cell is the membrane electrode assembly (MEA), which is composed of a solid polymer electrolyte (SPE) sandwiched between two electronically conductive electrodes, as shown in Figure 3.

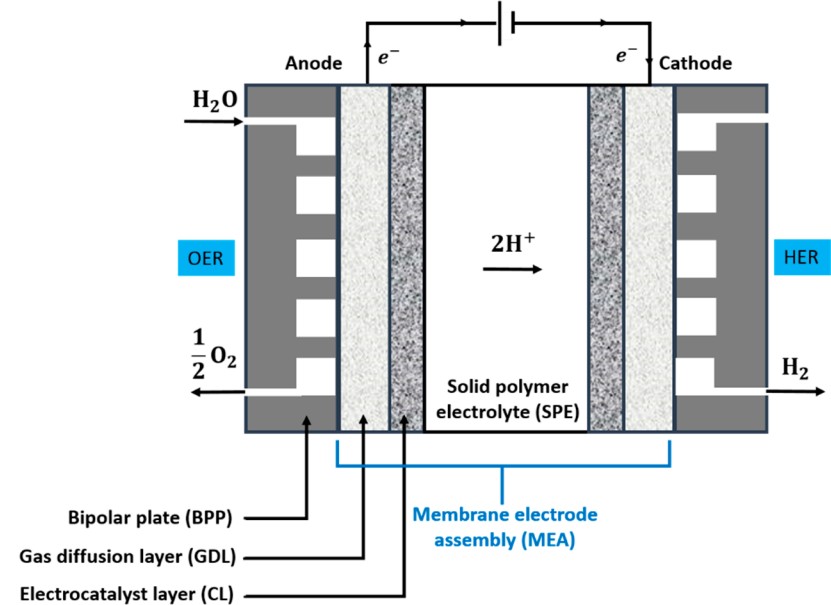

**Figure 3.** Basic, key components of a PEM WE.

The SPE must fulfil particular requirements, such as high chemical and mechanical stability, low gas permeability, and high proton conductivity. In this regard, Nafion® is the most commonly used polymer membrane due to high proton conductivity, good mechanical stability and acceptable gas crossover. The electrodes are composed of a porous catalyst layer (CL) with electrocatalysts dispersed on a nanoporous support to lower the activation energy and promote charge transfer kinetics. Next comes a more openly porous gas diffusion layer (GDL), also acting as a current collector alone or by the help of additional metallic meshes or sinters. These are finally encased by bipolar plates (BPPs) which direct and distribute gases in flow channels, separate the anode and cathode side environments, and connect a cathode electrically to the next anode [58].

Due to the acidic environment of the cell, the catalysts for the hydrogen evolution reactions (HER) on the cathode and the oxygen evolution reactions (OER) on the anode are essentially dependent on noble metals and their alloys. Pt nanoparticles on carbon support is by far the best catalyst material for the HER because of their good catalytic activity and high corrosion resistance. Besides, Pd and Ir nanoparticles supported on carbon materials are also commonly utilized as HER electrocatalysts [65]. Less expensive earth-abundant materials such as sulfides, phosphides, carbides and nitrides [18], cobalt clathrochelate [66], polyoxometallates [62] have been proposed as alternative HER catalysts.

The slowest reaction is normally the OER of the anode, determining the reaction rate of the overall process. Non-noble catalysts such as Ni and Co in contact with the acidic electrolyte would start to corrode, meanwhile the Pt surface will be covered by a low conducting oxide film, which reduces the catalytic activity for the OER. In this respect, Ir and Ru oxide-based catalysts are typically used for the OER because of their high structural stability. As reported by Ahn and Holze [67], Ru oxide appears to be the most catalytically active electrode with the smallest activation overpotential at 353 K, followed by Ir/Ru-oxide, Ir-oxide, Ir metal, Rh-oxide, Rh metal and Pt. However, Ir is scarce, its average mass fraction in crustal rock is only 0.001 ppm [54].

## 3. State-of-the-Art Devices

After General Electric developed the PEM WE technology, its application was mostly limited to oxygen production [68], e.g., for submarine and spacecraft applications. In the late 1980s, the first pressurized PEM electrolyzers for $H_2$ production up to 100 bar with efficient MEAs, were constructed and tested [69,70]. Since then, MEAs with Ir, Ru- and Pt-based electrocatalysts and Nafion$^{®}$ proton conductor polymer electrolyte have dominated frontier PEM electrolyzer cell designs [18,71].

The state-of-the-art OER catalyst for PEM electrolyzers is an oxide mixture composed of $Ru_2O$ and $IrO_2$ [72], e.g., $Ir_{0.7}Ru_{0.3}O_2$ [73] and $Ir_{0.4}Ru_{0.6}O_2$ [74], with slight differences in overpotential and stability when varying the composition of each oxide. Although $RuO_2$ has shown the best OER performance among all the other materials [54,74], its poor stability due to the corrosion [75] from the strong local acidity at the perfluorosulfonic membrane, and high anodic potential, it requires the addition of the more stable $IrO_2$ [76,77]. However, Ir is one of the rarest elements on earth, and this sets the requirement to reduce/replace the Ir content in order to lower the price, such as by adding other elements that are more earth abundant, e.g., Co [78], Ta [79], and Sn [80]. A recent study reported the state-of-the-art OER performance of fluoride doped $MnO_2$, $IrO_2$ solid solution $((Mn_{1-x}Ir_x)O_2:F)$, with even lower onset potential than $IrO_2$ [81], may further reduce the Ir loading of the OER catalysts.

For the cathode, it is established that Pt, especially highly dispersed C-based Pt, is the benchmark HER catalyst for PEM electrolyzer [18]. In fact, less research efforts have been made on the cathode material for PEM electrolyzers [54]. The reason is partially that the exchange current density of $H^+/H_2$ on Pt is almost 1000 times larger than that of $H_2O/O_2$ on Ir [82], and Ir is also more precious than Pt, therefore research has been mainly focused on how to reduce the cost and increase the efficiency of OER catalyst. However, as the cathode side also contributes to a large extent in the cost of a PEM electrolyzer, it is necessary and important to reduce the loading of Pt [83], or replace it with efficient earth abundant electrocatalysts, such as $MoS_2$ [84] or CoP [85]. This effort is briefly summarized below and as we set out earlier, our main target is to document actual application of EACs in full PEM WE cells.

Reported PEM electrolyzers with state-of-the-art electrocatalysts are summarized in Table 2. One can notice that the performance of a PEM electrolyzer is not only determined by the electrocatalysts, but also by other elements, e.g., operation temperature, cell area and membrane type. However, those elements are out of the scope of this review, hence they are not to be discussed here.

**Table 2.** PEM electrolyzers with state-of-the-art electrocatalysts.

| Cathode | Anode | T | Test Cell | Current Density | Cell Voltage | Ref. |
|---|---|---|---|---|---|---|
| Pt/C<br>0.5 mg$_{Pt}$/cm$^2$ | Ir$_{0.5}$Ru$_{0.3}$O$_2$<br>2.5 mg$_{oxide}$/cm$^2$ | 25 °C | 5 cm$^2$ PEM cell, Nafion 115 | 1 A/cm$^2$ | ~2.2 V | [86] |
| Pt/C<br>0.5 mg$_{Pt}$/cm$^2$ | Ir$_{0.7}$Ru$_{0.5}$O$_2$<br>2.5 mg$_{oxide}$/cm$^2$ | | | | ~2.3 V | |
| Pt/C<br>0.5 mg$_{Pt}$/cm$^2$ | Ir$_{0.7}$Ru$_{0.5}$O$_2$<br>1.5 mg$_{oxide}$/cm$^2$ | 90 °C | 5 cm$^2$ PEM cell, Nafion 115 | 2.6 A/cm$^2$ | 1.8 V | [73] |
| Pt/C<br>0.4 mg$_{Pt}$/cm$^2$ | Ir$_{0.7}$Ru$_{0.3}$O$_2$<br>thermally treated<br>1.0 mg$_{oxide}$/cm$^2$ | 80 °C | 25 cm$^2$ PEM cell, Nafion 212 CS | 1 A/cm$^2$ | ~1.7 V | [87] |
| Pt/C<br>0.1 mg$_{Pt}$/cm$^2$ | Ir$_{0.7}$Ru$_{0.3}$O$_2$<br>1.5 mg$_{oxide}$/cm$^2$ | 90 °C | 5 cm$^2$ PEM cell, Aquivion ionomer | 1.3 A/cm$^2$ | 1.6 V | [88] |
| Pt/C<br>0.4 mg$_{Pt}$/cm$^2$ | Ir$_{0.6}$Ru$_{0.4}$O$_2$<br>2.5 mg$_{oxide}$/cm$^2$ | 80 °C | 5 cm$^2$ PEM cell, Nafion 115 | 1 A/cm$^2$ | 1.567 V | [79] |
| Pt/C<br>0.4 mg$_{Pt}$/cm$^2$ | Ir$_{0.4}$Ru$_{0.6}$O$_2$<br>1.5 mg$_{oxide}$/cm$^2$ | 80 °C | 5 cm$^2$ PEM cell, Nafion 115 | 1 A/cm$^2$ | 1.676 V | [77] |
| Pt/C<br>0.5 mg$_{Pt}$/cm$^2$ | Ir$_{0.2}$Ru$_{0.8}$O$_2$<br>1.5 mg$_{oxide}$/cm$^2$ | 80 °C | 5 cm$^2$ PEM cell, Nafion® 1035 | 1 A/cm$^2$ | 1.622 V | [74] |

## 4. Earth-Abundant Cathode Materials

Thus far, we have explored the theory and principles of PEM WE and summarized the state-of-the-art devices demonstrated in the literature. In the following sections, we will explore the most promising earth-abundant electrocatalyst materials that have been used in full PEM WE cells, replacing noble metal-based anodes and cathodes, especially under acidic conditions.

### 4.1. Molybdenum Sulfide, MoS$_2$

Molybdenum sulfide (MoS$_2$)-based materials are among the most extensively studied materials as catalyst for HER over the past decade due to their excellent stability, high activity, earth abundancy and low price. MoS$_2$ exists in nature with an atomic structure resembling that of graphite, a layered structure where each layer consists of a molybdenum layer sandwiched between two sulfur layers. Alternatively, the monolayers can be described as consisting of either edge sharing trigonal prisms (2H) or octahedrons (1T). Packing of these layers gives the basis for the three polytypes of bulk MoS$_2$ (Figure 4).

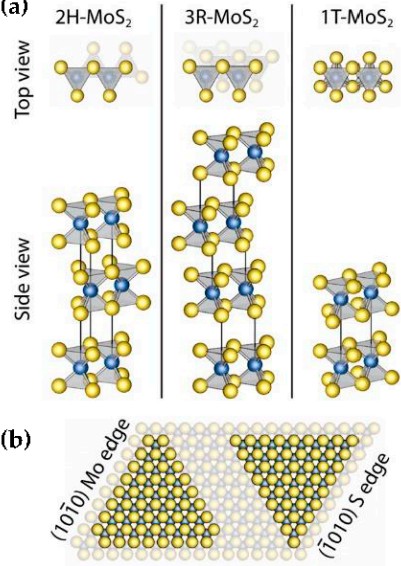

**Figure 4.** Figure showing the structures of MoS$_2$. (**a**) Illustration of the layer packing in the three polytypes: 2H, 3R and 1T. (**b**) Top view of MoS$_2$. Reprinted with permission from [89]. Copyright 2014 American Chemical Society.

Despite the early indications of low HER activity for bulk $MoS_2$ [90], molybdenum sulfides turned out to be promising for replacing Pt. Theoretical work by Hinnemann et al. in 2005 showed that the edges are in fact catalytically active [91]. Using Density Functional Theory (DFT) they calculated the hydrogen binding energy of the $Mo(\bar{1}010)$ edge, where sulphur is unsaturated, and found it to be close to ideal value of 0 eV [89]. In addition, they fabricated an MEA using Nafion®, nanoparticle $MoS_2$ on graphite as cathode, and Pt as anode, which achieved a current density of 10 mA/cm$^2$ at only 175 mV of overpotential. This was the best activity shown for an acid-stable and earth abundant catalyst at that time. Two years later, their theoretical prediction of the edges being the activity centers was confirmed experimentally by Jaramillo et al. [42]. They deposited monolayer $MoS_2$ on Au(111) with physical vapor deposition in an $H_2S$ environment. After finding total edge lengths with STM and comparing with catalytic activity for various samples, they found that the reaction rate scaled with particle perimeter and not area. These findings sparked an interest in improving the catalytic activity in $MoS_2$ that is still growing today.

Since the main objective of the present review is to review the literature on device-tested electrodes, we will not go deep into the vast literature on $MoS_2$-based electrocatalysts. We will rather briefly mention some of the methods that have been identified for increasing the HER activity of $MoS_2$. One of the first and obvious approaches was to maximize the edge sites by making small particles. This led to investigations of the activity of $[Mo_3S_4]^{4+}$-clusters that showed HER activity but were less stable [92]. Some years later, $[Mo_3S_{13}]^{2-}$-clusters became a hot topic after results showing one of the highest per site activities [33]. Another approach that has produced promising results is to deposit molybdenum sulfide onto something highly conducting and/or with high surface area, like nanotubes, nanowires, reduced graphene oxide etc. [93–96]. Depending on the methods used, one often ends up with amorphous $MoS_x$. Efforts to improve the activity of the semiconductor phase comprise doping, introducing vacancies, and strain engineering, which can activate the basal plane and edges that are not intrinsically active [97–100]. The 1T phase is metastable, however, the metallic nature makes it highly conductive compared to the 2H phase, and, in addition, the basal plane is active as well, resulting in promising HER activity [101,102]. For more in-depth reviews the reader is referred to a number of reviews [84,89,103,104]. Despite all these efforts to improve the catalytic properties over the past decade, there are, to the best of our knowledge, only the following few reports on molybdenum sulfide-based cathodes implemented in a PEM cell.

In 2014, Corrales-Sánchez et al. were the first to report the performance of a PEM cell using $MoS_2$-based cathodes [84]. They reported the performance of three different types of $MoS_2$-based electrodes, bare pristine $MoS_2$, $MoS_2$ mixed with commercial conductive carbon, Vulcan® XC72, and $MoS_2$ nanoparticles on reduced graphene oxide. The MEA used in the PEM cell consisted of $IrO_2$ particles and anode material that was spray deposited on each side of a Nafion membrane. Porous titanium diffusion layer and titanium current collectors on both sides of the MEA were sandwiched by the cell housing. The pristine $MoS_2$ was the worst performing cathode investigated achieving a current density of approximately 0.02 A/cm$^2$ at 1.9 V. Their best performing $MoS_2$/rGO electrode achieved a current density of 0.1 A/cm$^2$, while the best mixture of $MoS_2$ and Vulcan® (47 wt.% $MoS_2$) reached almost 0.3 A/cm$^2$ at 1.9 V in the initial test. The latter electrode went through a stability test for 18 h at 2.0 V. The current density actually increased steadily for 15 h and reached 0.35 A/cm$^2$. The authors speculated that the increase might be due to hydration effects. Furthermore, they also tested the effect of hot pressing of the MEA, which is recommended to ensure good contact between electrode and membrane. For three different $MoS_2$/Vulcan mixes, the unpressed MEAs performed better than the hot-pressed ones.

Ng et al. identified three types of Mo-based cathode materials with excellent HER activity from three electrode measurements in 2015 [105]. They later loaded the materials onto carbon black and tested them as cathodes in a PEM electrolyzer with Nafion as membrane and Ir on Ti-mesh as anode. One of their electrodes was based on molybdenum sulfide with an excess of sulfur according to the XPS measurement. The electrode exhibited a good performance and required 1.86 V to reach 0.5 A/cm$^2$

in addition to good stability. Furthermore, the current density reached over 0.9 A/cm$^2$ at 2 V. Another cathode, based on Mo$_3$S$_{13}$ clusters, required only 1.81 V to reach 0.5 A/cm$^2$, while at 2 V the current density reached almost 1.1 A/cm$^2$. In the stability test, however, the current density dropped by approximately 120 mA/cm$^2$ over a period of 14 h at 1.85 V most likely due to detachment from the support or degradation of the clusters. The third and last material they tested was based on sulfur doped molybdenum phosphide and performed slightly better than the Mo$_3$S$_{13}$ electrode. These are the best performances reported for molybdenum sulfide cathode in PEM electrolyzers to this day.

In early 2016 Kumar et al. reported that a cell with a MoS$_2$ nanocapsule cathode maintained a current density of approximately 60 mA/cm$^2$ for 200 h at 2.0 V [106]. The cell consisted of a Nafion membrane and IrO$_2$ anode. The low performance is likely due to low conductivity and is comparable to that reported for bare MoS$_2$ [84]. A study of this system mixed with carbon black should follow to allow comparison with other systems reviewed here.

The same year, Lu et al. reported the performance of an electrolyzer using amorphous molybdenum sulfide coated on a carbon cloth as cathode [107]. The cathode was synthesized by using thermolysis to form amorphous MoS$_x$ on the carbon cloth. A post treatment with remote H$_2$ plasma introduced sulfur vacancies. The cell consisted of a Nafion membrane and RuO$_2$ nanoparticles on carbon paper as the anode. The cell required 2.76 V to reach 1 A/cm$^2$ and the current density at 2.0 V was slightly above 0.3 A/cm$^2$. Earlier this year, Kim et al. published work on a similar cathode. They deposited amorphous molybdenum sulfide on carbon paper using electrodeposition. The PEM cell used a Nafion membrane and electrodeposited IrO$_2$ on carbon paper as anode. They investigated the effect of deposition potential and time on the performance. The best performing electrode reached a current 0.37 A/cm$^2$ at 1.9 V [108].

### 4.2. Nickel Phosphide, Ni$_2$P

Nickel phosphide (Ni$_2$P) has been demonstrated as one of the best earth-abundant electrocatalysts for HER [34,109]. Extensive investigations on Ni$_2$P have been performed in a three-electrode electrochemical cell and Ni$_2$P exhibits the superior activity to split water with low overpotentials, while sustaining high current densities [110–115]. However, after a thorough literature review, there are no reports, to our best of knowledge, that have implemented Ni$_2$P in a PEM device. Nevertheless, we compare Ni$_2$P with other earth-abundant electrocatalysts, and the recent developments on Ni$_2$P as electrocatalysts for HER are briefly reviewed.

Ni$_2$P can be synthesized by a variety of methods including solution-phase synthesis and gas-solid synthesis. The solution-phase synthesis is performed by using tri-n-octylphosphine (TOP) as a phosphorus source to react with Ni precursor [116]. At elevated temperatures (above 300 °C), the TOP vaporizes rapidly and then phosphorizes different precursors, such as bulk Ni or Ni thin films, by forming Ni$_2$P. For instance, Read et al. successfully synthesized Ni$_2$P thin film on Ni substrate by the solution-phase synthesis method [113]. Figure 5a shows SEM images of representative Ni$_2$P film formed on the surface of Ni foil and the resulting Ni$_2$P is highly porous. The corresponding powder XRD pattern in Figure 5c, clearly shows that both Ni$_2$P and Ni are present without other impurities. The EDS element maps in Figures 5d and 2e further confirm the presence of Ni and P at the surface and the existence of a sharp interface between the Ni$_2$P coating and the underlying Ni substrate. Figure 5f shows polarization data for the HER in 0.5 M H$_2$SO$_4$ for a few transition metal phosphides (Ni$_2$P, Fe$_2$P, Co$_2$P, Ni$_2$P, Cu$_3$P, and NiFeP) as cathodes. Ni$_2$P showed the best HER performance in acidic solutions among those and required overpotentials of only $-128$ mV and $-153$ mV to reach a current density of $-10$ mA/cm$^2$ and $-20$ mA/cm$^2$, respectively. However, in alkaline media, all tested metal phosphide electrodes exhibit lower electrocatalytic HER activity compared to those in acidic conditions. Ni$_2$P films require overpotentials of around $-200$ mV to reach current densities of $-10$ mA/cm$^2$ in 1.0 M KOH.

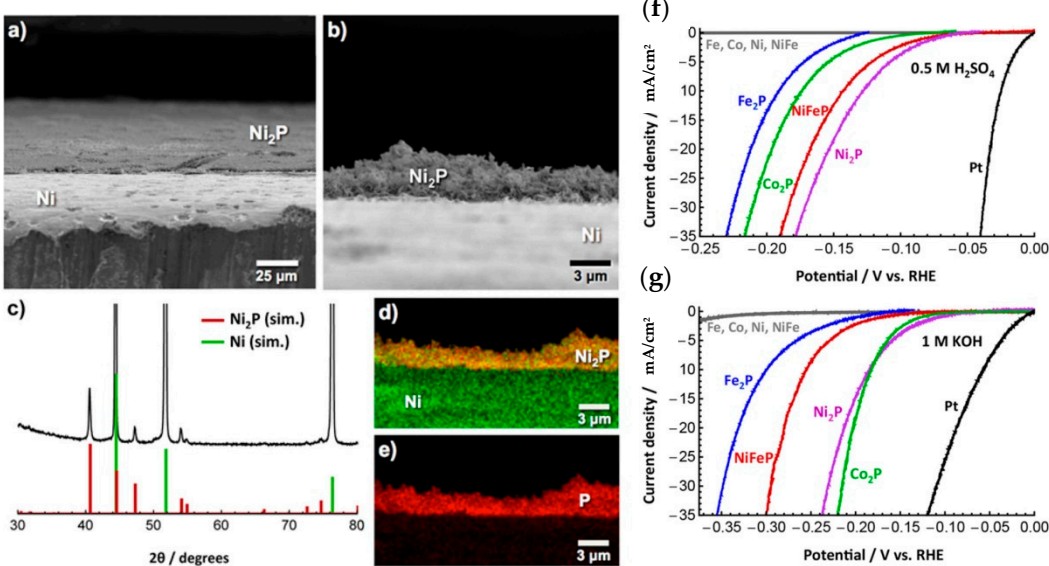

**Figure 5.** (**a**,**b**) SEM images of a representative $Ni_2P$ film on Ni. (**c**) Experimental powder XRD pattern of a Ni2P sample (black), with the simulated (sim.) patterns of Ni (green) and $Ni_2P$ (red) shown for comparison. The y-axis was truncated to highlight the $Ni_2P$ as the Ni signal would otherwise dominate. (**d**,**e**) EDS elemental maps of a cross-section of the sample showing the presence of both Ni (green) and P (red) in a 2:1 ratio. (**f**) Polarization data for the HER in 0.5 M $H_2SO_4$ and (**g**) 1 M KOH for a series of metal phosphide films, along with a Pt mesh electrode for comparison. Reprinted with permission from [113]. Copyright 2017 The Royal Society of Chemistry.

Gas-solid synthesis has also been implemented to synthesize $Ni_2P$, where hypophosphites, for instance $NH_4H_2PO_2$ and $NaH_2PO_2$, can decompose and release $PH_3$ at elevated temperatures;

$$2NaH_2PO_2 \rightarrow PH_3 + Na_2HPO_4 \tag{6}$$

The $PH_3$ can further react directly with Ni precursors, such as metal oxides and metal hydroxides, to form $Ni_2P$ [117–121]. For instance, Sun et al., reported one porous multishelled $Ni_2P$, which was successfully synthesized by a gas-solid method [120]. The porous multishelled NiO precursor was reacted into $Ni_2P$ by using $NaH_2PO_2$ as the phosphorus source, as shown in Figure 6a. Electrochemical measurements were performed in a 1 M KOH solution. Figure 6b shows the linear sweep curves for carbon, nanostructured $Ni_2P$, hierarchical $Ni_2P$, multishelled $Ni_2P$, and Pt/C. The multishelled $Ni_2P$ exhibits a small overpotential of 10 mV (at current density of 1.0 mA/cm$^2$) and a rapid cathodic current increase as more negative potentials were applied. The overpotential driving a cathodic current density of 10 mA/cm$^2$ was 98 mV, which is much lower than that observed on hierarchical $Ni_2P$ (298 mV) and nanostructured $Ni_2P$ (214 mV). Figure 6c shows the Tafel plots of the tested samples. At lower overpotentials, Tafel analysis on the multishelled $Ni_2P$ exhibits a slope of 86.4 mV/decade, which is much smaller than those of hierarchical $Ni_2P$ (108.4 mV/decade) and nanostructured $Ni_2P$ (125.4 mV/decade), suggesting faster HER kinetics of the multishelled $Ni_2P$. At the high-overpotential regime, a slightly upward deviation is observed in Tafel plots of Pt/C and hierarchical $Ni_2P$, which could stem from the rate-limiting step gradually changing from the Heyrovsky to the Volmer mechanism at high current densities [122]. This porous multishelled structure endows $Ni_2P$ with short charge transport distances and abundant active sites, resulting in superior catalytic activity than those of $Ni_2P$ with other morphologies [120].

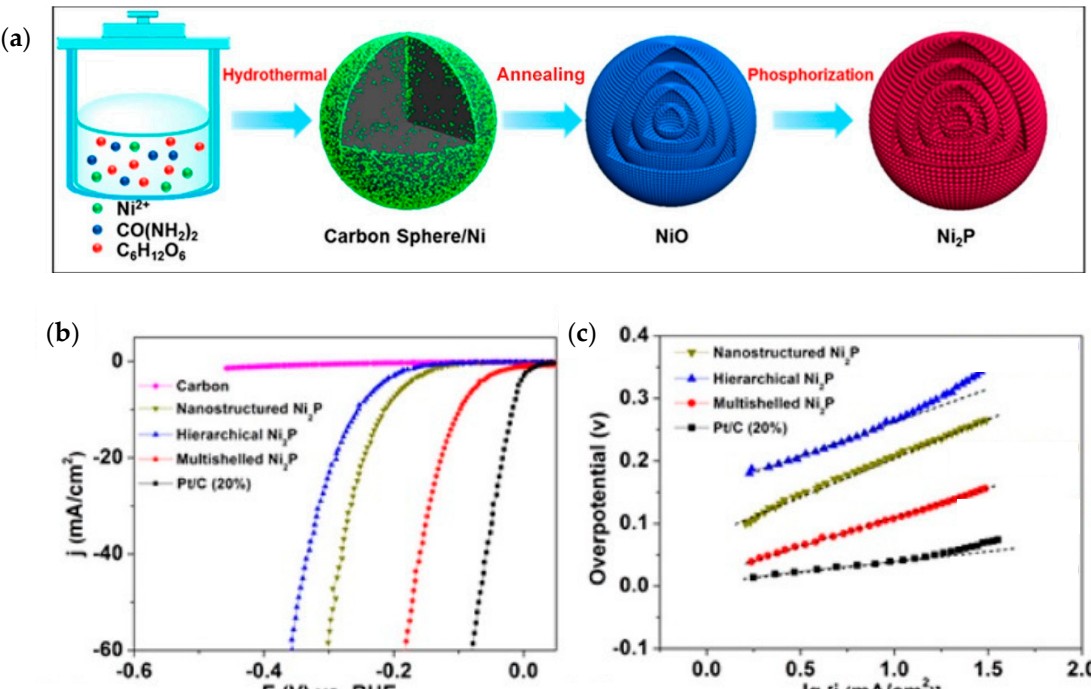

**Figure 6.** (**a**) Synthetic schematic illustration and material characterization of the multishelled $Ni_2P$. (**b**) Linear sweep voltammetry (LSV) polarization curves of bare carbon, nanostructured $Ni_2P$, hierarchical $Ni_2P$, multishelled $Ni_2P$, and benchmark Pt/C in 1 M KOH at a scan rate of 5 mV s$^{-1}$. (**c**) Corresponding Tafel plots with linear fittings. Reprinted with permission from [120]. Copyright 2017 American Chemical Society.

A catalytic reaction is highly sensitive to the surface of the catalyst. One of the most common strategies to enhance the catalyst performance is by increasing the active facet of the catalyst. Several computational studies have suggested that $Ni_2P(001)$ surface is an active facet for HER due to an ensemble effect, whereby the presence of P decreases the number of metal-hollow sites, providing a relatively weak binding between protons and Ni−P bridges, the sites to facilitate catalysis of the HER [123,124]. Later on, Popczun et al. successfully synthesized $Ni_2P$ nanoparticles which possessed a high density of exposed (001) facets (as shown in Figure 7) and then these $Ni_2P$ were tested as cathodes for the HER in 0.50 M $H_2SO_4$ [125]. The overpotentials required for the $Ni_2P$ nanoparticle to produce cathodic current densities of 20 and 100 mA/cm$^2$ were 130 and 180 mV, respectively. These overpotentials are lower than those of none-preferred facet $Ni_2P$ [113] and other non-Pt HER electrocatalysts, including bulk $MoS_2$ [94] and MoC [126]. Figure 7c displays corresponding Tafel plots for $Ni_2P$ electrodes. Tafel analyses of the $Ni_2P$ nanoparticles show an exchange current density of $3.3 \times 10^{-5}$ A/cm$^2$ and a Tafel slope of ~46 mV/decade in the overpotential region of 25–125 mV. At higher overpotentials (150–200 mV), the Tafel slope and exchange current density increased to ~81 mV/decade and $4.9 \times 10^{-4}$ A/cm$^2$, respectively. Again, this Tafel slope behavior reflects the change in the rate-limiting step of the HER [122].

Cation doping is an effective strategy to improve the HER activity of electrocatalysts. A few cations, such as Mn, Fe and Mo, have been reported to dope $Ni_2P$ [110,111,127–129]. For instance, Li et al. synthesized a series of $(Ni_xFe_{1-x})_2P$ by varying the amount of Fe doping ratio [128]. They found out that HER activities for $(Ni_xFe_{1-x})_2P$ electrodes show a volcano shape as a function of Fe doping ratio (see Figure 8); HER activities first increased as Fe content increased until the composition reaches $(Ni_{0.33}Fe_{0.67})_2P$. Then, by further increasing the Fe content, HER performance decreased gradually. $(Ni_{0.33}Fe_{0.67})_2P$ shows the best performance among the tested $(Ni_xFe_{1-x})_2P$ samples, with a small overpotential of 214 mV to reach cathodic current densities of 50 mA/cm$^2$. Such

an interesting behavior could stem from an increase in the electrocatalytically active surface areas, as well as a change in the electronic structure with increasing Fe content [128,130].

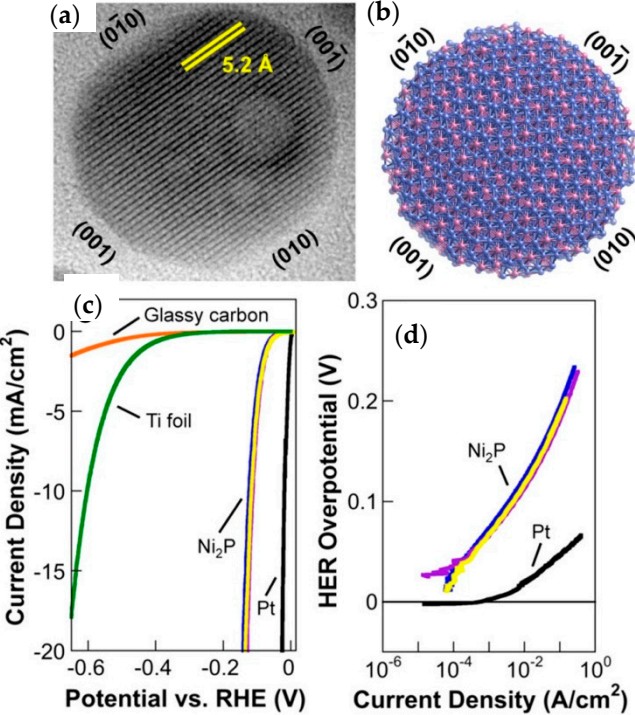

**Figure 7.** (**a**) High resolution TEM image of a representative $Ni_2P$ nanoparticle, highlighting the exposed $Ni_2P$ (001) facet and the 5.2 Å lattice fringes that correspond to the (010) planes. (**b**) Proposed structural model of the $Ni_2P$ nanoparticles. (**c**) Polarization data for three individual $Ni_2P$ electrodes in 0.5 M $H_2SO_4$, along with glassy carbon, Ti foil, and Pt in 0.5 M $H_2SO_4$, for comparison. (**d**) Corresponding Tafel plots for the $Ni_2P$ and Pt electrodes. Reprinted with permission from [125]. Copyright 2013 American Chemical Society.

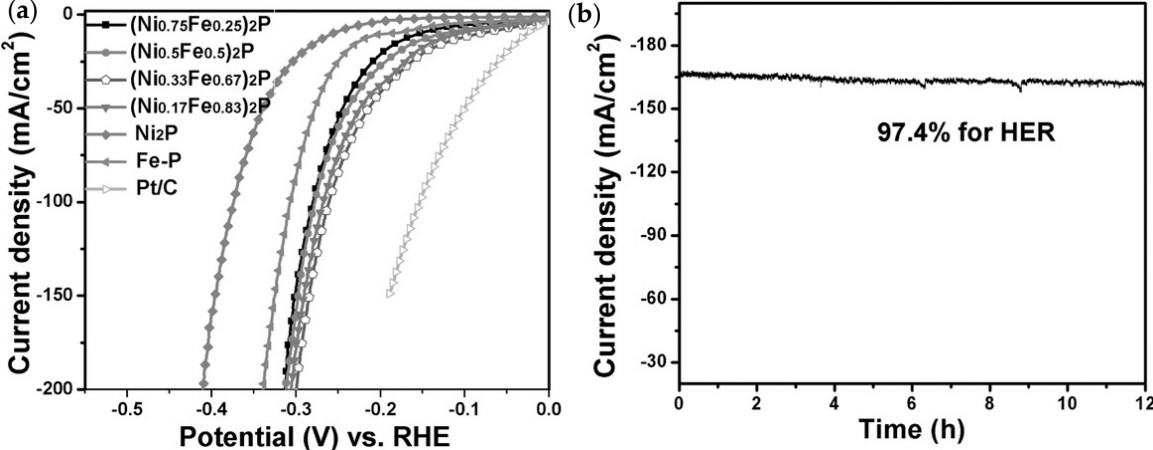

**Figure 8.** (**a**) Polarization curves of a series of $(Ni_xFe_{1-x})_2P$ and commercial Pt/C electrodes for HER at a scan rate of 5 mV/s. (**b**) Time-dependent current density curve of $(Ni_{0.33}Fe_{0.67})_2P$ at a constant overpotential of ≈285 mV. Reprinted with permission from [128]. Copyright 2017 WILEY-VCH Verlag GmbH & Co. KGaA, Weinheim.

Electron conductivity and dispersion of electrocatalysts also severely affect the catalytic activity of the electrocatalysts. Various carbon materials, such as carbon nanotube and carbon cloth, which possess both strong electronic conductivity and high surface area, have been implemented as $Ni_2P$

support materials to enhance HER activity [131–141]. For instance, Pan et al. reported a hybrid material where $Ni_2P$ was supported on multiwalled carbon nanotubes ($Ni_2P/CNT$), as shown in Figure 9a [136]. The HER catalytic activity of the $Ni_2P/CNT$ nanohybrid was evaluated in 0.5 M $H_2SO_4$. $Ni_2P/CNT$ exhibits high catalytic activity with a low overpotential of 124 mV when current density reached 10 mA/cm$^2$. The corresponding Tafel slope is 53 mV/decade, reflecting that the HER reaction took place via a fast Volmer step followed by a rate-determining Heyrovsky step [142]. Furthermore, the turnover frequency (TOF) was calculated and normalized by the total number of active sites. To achieve a TOF value of 0.1 s$^{-1}$, $Ni_2P/CNT$ needs only an overpotential of about 170 mV, much smaller than that required by the $Ni_{12}P_5/CNT$ and $Ni/CNT$ hybrid materials, further showcasing the high catalytic activity of $Ni_2P$.

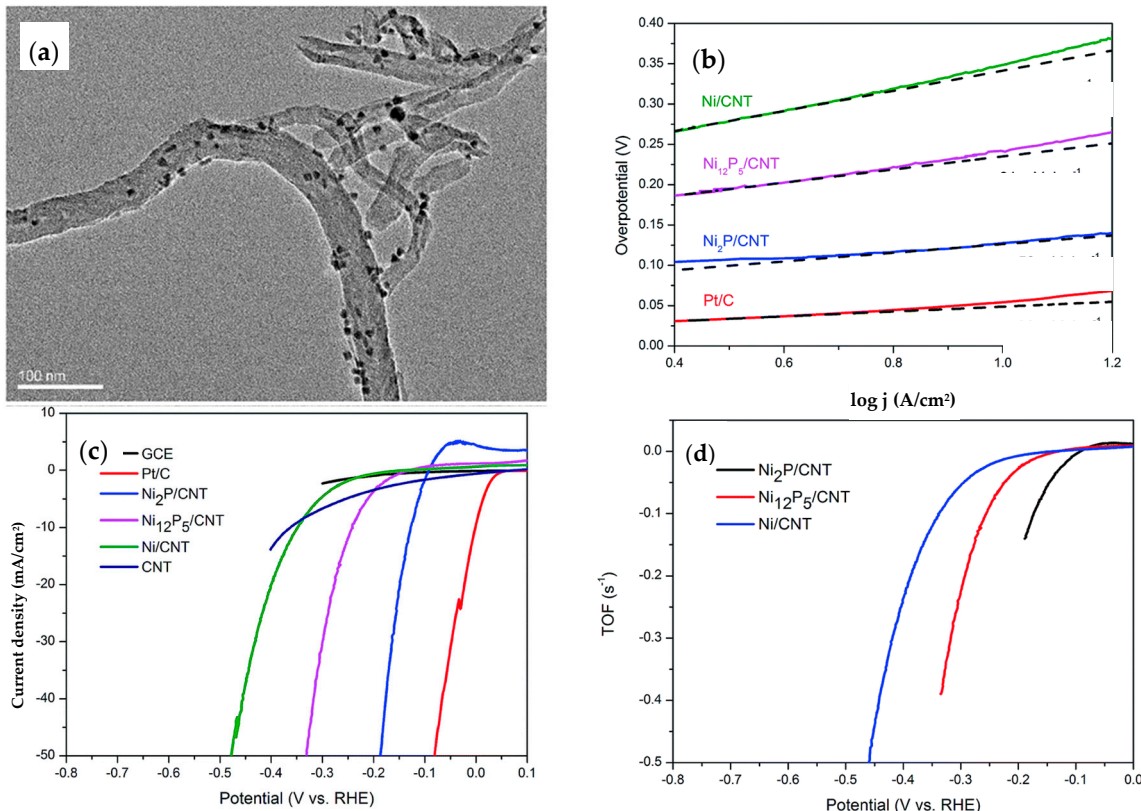

**Figure 9.** (**a**) TEM image of $Ni_2P/CNT$. (**b**) Tafel plots of the $Ni_2P/CNT$, $Ni_{12}P_5/CNT$, $Ni/CNT$ and Pt/C. (**c**) LSV curves of the $Ni_2P/CNT$, $Ni_{12}P_5/CNT$, $Ni/CNT$, Pt/C, CNT and bare GCE in 0.5 M $H_2SO_4$ with a scan rate of 5 mV/s. (**d**) Calculated TOFs for the $Ni_2P/CNT$, $Ni_{12}P_5/CNT$ and $Ni/CNT$ in 0.5 M $H_2SO_4$. Reprinted with permission from [136]. Copyright 2017 The Royal Society of Chemistry.

### 4.3. Iron Sulfides, $Fe_xS_y$

Metal chalcogenides have received interest as HER electrocatalysts over the past decades such as molybdenum sulfide $MoS_2$ [42], tungsten sulfide $WS_2$ [143], iron phosphide FeP [47] or nickel phosphide $Ni_2P$ [125]. Among them, iron sulfides (generally noted as $Fe_xS_y$) show great interest, especially being the most abundant mineral on the Earth's surface, and pyrrhotite $Fe_9S_{10}$ being the most abundant iron sulfide in the Earth and solar system [144,145].

To our knowledge, the only study of iron sulfide electrocatalysts in a PEM WE device has been published by Di Giovanni et al. [145]. In this paper the authors describe the synthesis and characterization of different stoichiometries of iron sulfide $Fe_xS_y$ nanomaterials and their activity toward the HER. Pyrite $FeS_2$, greigite $Fe_3S_4$, and pyrrhotite $Fe_9S_{10}$ crystalline phases were first prepared using a polyol synthetic route. Morphological and electronic properties of the prepared nanoparticles were characterized, as well as their electrochemical properties. Greigite is formed of micrometer-sized

gypsum flowerlike particles consisting of thin platelets with a very high aspect ratio. Pyrite particles have a hierarchical morphology consisting of large micrometer-sized spheres of aggregated smaller particles. Their performances were investigated in situ in a PEM electrolyzer single cell. MEA were prepared using pyrite, pyrrhotite, or greigite as the anode catalyst and tested in a PEM electrolysis single cell. The catalysts were not supported, but were mixed with 20% of carbon black. Nafion 115 (125 μm) was used as the membrane and $IrO_2$ as the anode catalyst. A cross section SEM image is presented in Figure 10 left. For the same catalyst loading, both ex situ and in situ (Figure 10 right) electrochemical experiments showed that pyrite ($FeS_2$) is the most active compared to greigite $Fe_3S_4$ and pyrrhotite $Fe_9S_{10}$, with the electrocatalysis starting at an overpotential of ca. 180 mV. These three materials exhibited a very stable behavior during measurement, with no activity degradation for at least 5 days. All catalysts have been tested in a PEM electrolysis single cell, and pyrite $FeS_2$ allows a current density of 2 A/cm$^2$ at a voltage of 2.3 V.

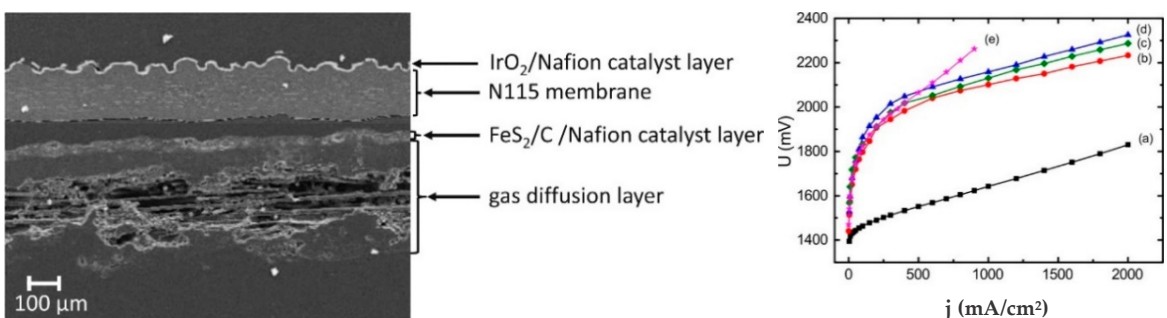

**Figure 10.** (**left**) SEM image of the cross section of the MEA $IrO_2$/Nafion/pyrite $FeS_2$. (**right**) Polarization curves at 80 °C and atmospheric pressure with (a) Pt/C-based MEA (black squares), (b) pyrite-based MEA (red dots), (c) greigite-based MEA (green diamonds), (d) pyrrhotite-based MEA (blue triangles) and (e) selected carbon-only-based MEA (magenta stars). Reprinted with permission from [145]. Copyright 2018 American Chemical Society.

It is noteworthy that $Fe_xS_y$ based materials have been studied as electrocatalysts for the HER and showed promising results.

FeS pyrrhotite has been prepared by a solvothermal route and showed hexagonal shaped nanoparticles with size ranging from 50 to 500 nm, achieving electrocatalysis for molecular hydrogen evolution with no structural decomposition or activity decrease for at least six days at an overpotential of 350 mV in neutral water [146].

$Fe_2S$ pyrite has been prepared by Faber et al. by electron-beam evaporation on borosilicate substrates following by a thermal sulfidation [147]. The cathodic overpotential to drive the HER at 1 mA/cm$^2$ for $Fe_2S$ pyrite was 217 mV.

Miao et al. prepared mesoporous $Fe_2S$ materials with high surface area by a sol-gel method followed by a sulfurization treatment in an $H_2S$ atmosphere [148]. An interesting HER catalytic performance was achieved with a rather low overpotential of 96 mV at a current density of 10 mA/cm$^2$ and a Tafel slope of 78 mV/decade under alkaline conditions (pH 13).

Jasion et al. proposed the synthesis of nanostructured $Fe_2S$ [149]. By changing the Fe:S ratio in the precursor solution, they were able to preferentially synthesize either 1D wire or 2D disc nanostructures. The HER electrocatalytic activity of the nanostructured $FeS_2$ (drop-casted on a glassy carbon electrode) was measured via linear sweep voltammetry (LSV) and showed the best results for the 2D disc structures with an overpotential of just 50 mV larger than that of Pt.

Chua and Pumera investigated the electrochemical hydrogen evolution of natural $FeS_2$ [150]. Interestingly, they focused on the susceptibility of natural $FeS_2$ hydrogen evolution performances towards sulfide poisoning, a major issue for cathodic hydrogen evolution. The results showed a better response of the $FeS_2$ electrodes than platinum.

A hybrid catalyst of cobalt-doped $FeS_2$ nanosheets–carbon nanotubes for the HER was proposed by Wang et al. [151]. The pyrite phase of $Fe_{1-x}Co_xS_2$/CNT showed a low overpotential of ~120 mV at 20 mA/cm$^2$, a low Tafel slope of ~46 mV/decade, and long-term durability over 40 h of HER operation. Huang et al. employed carbon black as a support to prepare a cobalt-doped iron sulfide electrocatalyst with high-electrical conductivity and maximal active sites [152]. Electrochemical results showed an enhancement in the HER activity of Co-doped $FeS_2$ in comparison to undoped $FeS_2$ in acidic electrolyte (pH = 0). The overpotential necessary to drive a current density of 10 mA/cm$^2$ is 150 mV and only decreases by 1 mV after 500 cycles during a durability test.

Bi-functional iron-only electrocatalysts for both water splitting half reactions are proposed by Martindale et al. [153]. Full water splitting at a current density of 10 mA/cm$^2$ is achieved at a bias of ca. 2 V, which is stable for at least 3 days.

Iron sulfide alloys have also shown potential catalytic activity. Yu et al. report the 3D ternary nickel iron sulfide ($Ni_{0.7}Fe_{0.3}S_2$) microflowers with a hierarchically porous structure delivering an overpotential of 198 mV at a current density of 10 mA/cm$^2$ [154]. Zhu et al. proposed bimetallic iron-nickel sulfide ($Fe_{11.1\%}$–$Ni_3S_2$) nanoarrays supported on nickel foam having a $\eta_{10}$ of 126 mV [155].

A patent has also been filed for the use of iron sulfide in an electrolytic cell [156].

### 4.4. Carbon-Based Materials

Due to the earth abundance and high electronic conductivity, carbon based materials, such as carbon nanoparticles (CNPs), carbon nanotubes (CNTs), graphene, etc., are mostly used as the supporting material for the electron transfer between the substrates and the electrocatalysts [157]. One of the most successful carbon material used as electrocatalyst support is carbon black, which is a commercially available product with high surface area (ca. 200–1000 m$^2$/g) [158]. By uniformly dispersing electrocatalyst NPs on carbon black, the electrochemically active surface area (EASA) of the electrocatalyst can be maximized, and the amount of the catalyst, such as Pt, can be minimized. Pt/C is actually the benchmark HER catalyst for PEM electrolysis [159].

In order to further reduce the cost of $H_2$ produced by the PEM electrolyzer, other carbon-supported electrocatalysts, especially those only consist of earth abundant elements, such as $Mo_2C$/CNTs [126], A-Ni-C (atomically isolated Ni anchored on graphitic carbon) [160], Co-doped $FeS_2$/CNTs [151], CoFe nanoalloys encapsulated in N-doped graphene [161], $Ni_2P$/CNTs [136], $WO_2$/C nanowires [162], etc., have been studied as potential HER catalysts alternative to Pt. However, carbon-supported and Pt-free HER catalysts that have actually been tested in a real PEM device are rarely reported, and only a few can be found in the literature, and they are summarized in Table 3.

Nevertheless, the usage of C-based materials is not only limited to the anode. A recent study shows that carbon nitride ($C_3N_4$) resist the harsh conditions at the anode side and therefore can be used as the supporting material for OER catalysts, such as $IrO_2$, hence to reduce the Ir content at the anode [163].

### 4.5. Co-Clathrochelates

The interest in Co-clathrochelates as electrocatalysts is prompted by their ability to maintain the same ligand environment for Co in different oxidation states [164]. However, only a few studies can be found implementing Co-clathrochelates in PEM electrolyzers. As can be seen from Table 3, the cell performance when cathodes are impregnated with such stable Co-containing electrocatalyst complexes is comparable to other earth-abundant catalyst systems, achieving current densities of 0.65 and 1 A/cm$^2$ at 1.7 and 2.15 V, respectively (Dinh Nguyen et al. [165] and Grigoriev et al. [166]). In both these works, the Co-clathrochelates were implemented in 7 cm$^2$ cells, but with different loadings. Figure 11 left shows how a clean glassy carbon electrode (GCE) (a) is improved by addition of $[Co(dmg)_3(BF)_2]BF_4$ (a) and $Co(dmgBF_2)_2$ (b) in a 0.5 M $H_2SO_4$ aqueous solution. The two Co clathrochelate molecules are shown in Figure 11 right.

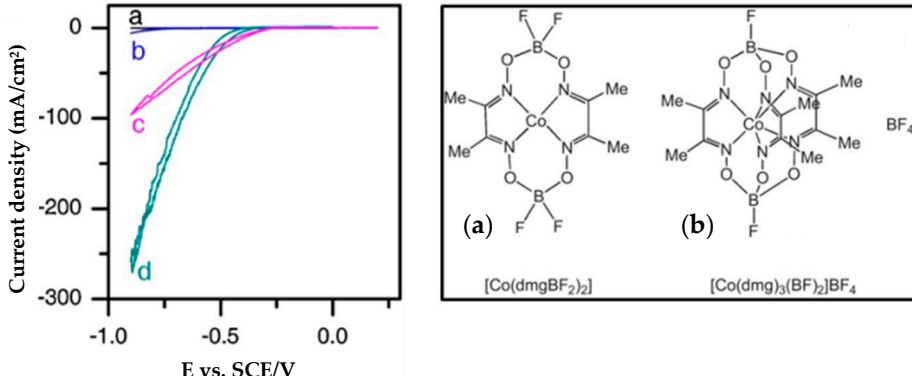

**Figure 11.** (**left**) Current-potential relations of (a) a clean glassy carbon electrode (GCE), (b) GCE modified with carbon black (Vulcan XC72) and Nafion 117, (c) GCE modified with Vulcan XC72 (70 wt.%), [Co(dmg)$_3$(BF)$_2$]BF$_4$ (30 wt.%) and Nafion 117, (d) GCE modified with Vulcan XC72 (70 wt.%), Co(dmgBF$_2$)$_2$ (30 wt.%) and Nafion 117, all in a 0.5 M H$_2$SO$_4$ aqueous solution, scan rate: 10 mV/s. (**right**) Molecular structure of the two Co clathrochelates. Reprinted with permission from [165]. Copyright 2012 Elsevier.

In Figure 11 left, the Co(dmgBF$_2$)$_2$ shows better electrochemical performance than [Co(dmg)$_3$(BF)$_2$]BF$_4$ in the three-electrode configuration. However, when the two electrode modifications above were implemented in single cells for i-V characterization and stability testing under operational conditions, the [Co(dmg)$_3$(BF)$_2$]BF$_4$ catalyst shows the best performance. The results are given in Figure 12 for current-potential and stability, respectively. The results reveal an increased cell voltage of 0.2–0.25 V when substituting the HER catalyst from Pt to Co-clathrochelates. The catalysts show no sign of degradation after 60 h of operation at 0.2 A/cm$^2$.

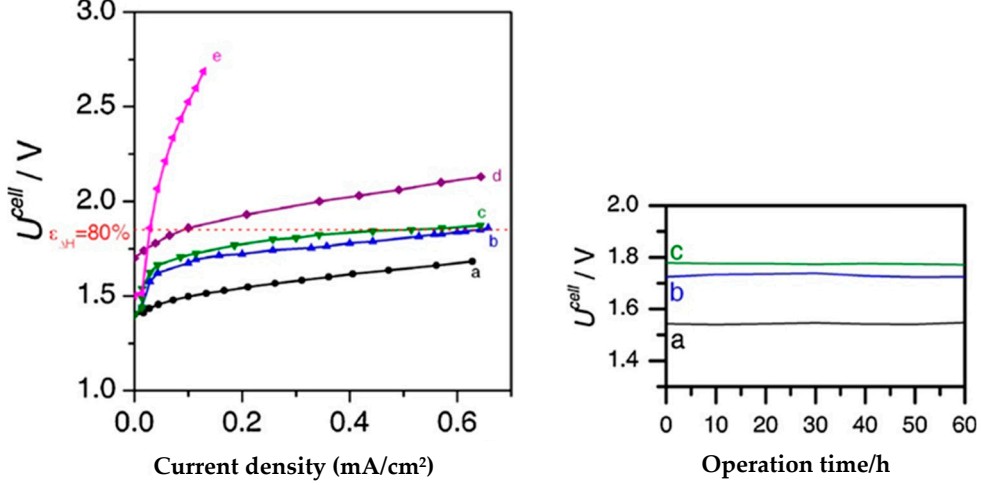

**Figure 12.** (**left**) Current-voltage performances for a 7 cm$^2$ single cell with different MEAs: (a) Ir(O$_2$)/Nafion 117/Pt(H$_2$), (b) Ir/Nafion 117/[Co(dmg)$_3$(BF)$_2$]BF$^4$-Vulcan XC72, (c) Ir/Nafion 117/[Co(dmgBF$_2$)$_2$]-Vulcan XC72, (d) Pt/Nafion 117/Pt, (e) Ir/Nafion 117/[Co(acac)$_3$]-Vulcan XC72. Experiments were carried at 60° and P = 1 atm. (**right**) Stability of the cells at 0.2 A/cm$^2$. Reprinted with permission from [165]. Copyright 2012 Elsevier.

In an earlier study by Millet et al. [62], a 23 cm$^2$ cell was prepared with a Co(dmgBF2)2 catalyst with a loading of 1 mg/cm$^2$ in carbon black as compared to the 2.5 mg/cm$^2$ of the same catalyst in in the study of Dinh Nguyen et al., resulting in an increased cell voltage of 0.1 V to obtain a current density of 0.5 A/cm$^2$. It must here be noted, however, that Miller's cell ran at 90 °C, while Dinh Nguyen's was operated at 60 °C, so the real difference under equal condition can be expected to be somewhat larger, suggesting that a 150% increase in the catalyst loading makes a significant impact on

cell efficiency. The discrepancy between the results in half-cell and full-cell testing clearly underlines the need for testing in operation conditions before concluding on electrochemical performance. Co- and Fe-hexachloroclathrochelates have also been applied by Grigoriev et al. in a full cell, impregnated on Vulcan XC72 Gas Diffusion Electrodes (GDEs) with a surface area of 7 cm$^2$ [166]. The main outcome is that substituting Co with Fe improves the electrocatalytic performance of the same macromolecule (Figure 13). One can also see that the overvoltage is around 0.25 V higher for the hexachloroclathrochelates than for the carbon supported Pt cathode used as reference. Comparing the results of Grigoriev et al. to the results reported by Dinh Nguyen et al. is difficult, since no information is given with respect to ohmic contributions to cell resistance for the former, while ohmic contributions are subtracted for the latter. However, the same difference in overvoltage can be seen with respect to carbon supported Pt.

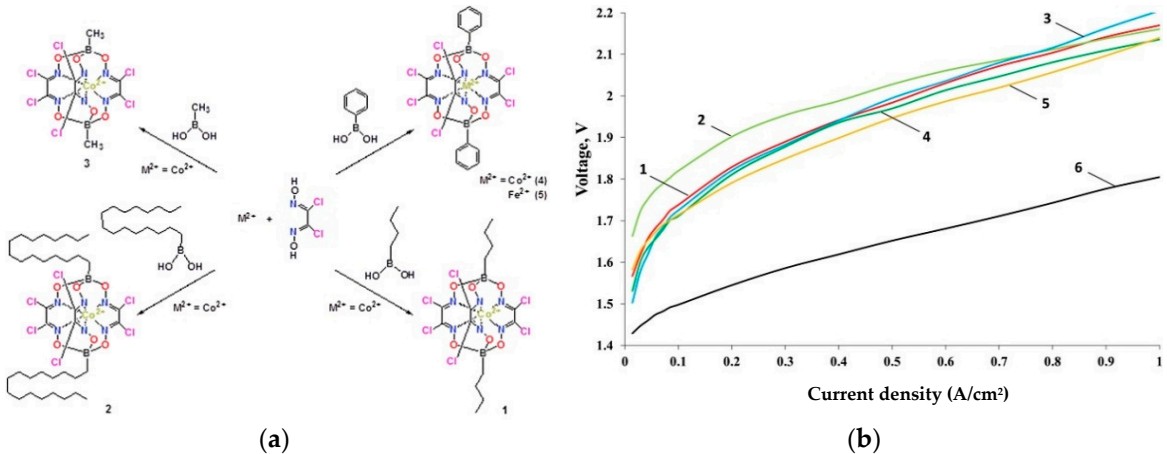

|     |     |
| :-: | :-: |
| (**a**) | (**b**) |

**Figure 13.** (**a**) Synthesis of the metal (II) hexachloroclthrochelates. (**b**) Current-voltage performances of MEAs with cathodes based on metal (II) clathrochelates Co(Cl$_2$Gm)$_3$(Bn-C$_4$H$_9$)$_2$ (1), Co(Cl$_2$Gm)$_3$(Bn-C$_{16}$H$_{33}$)$_2$ (2), Co(Cl$_2$Gm)$_3$(BCH$_3$)$_2$ (3), Co(Cl$_2$Gm)$_3$(BC$_6$H$_5$)$_2$ (4) and Fe(Cl$_2$Gm)$_3$(BC$_6$H$_5$)$_2$ (5) and Pt/Vulcan XC72 (6). Reprinted with permission from [166]. Copyright 2017 Elsevier.

Grigoriev et al. reported that the HER performance of Co-encapsulating macromolecules is improved by adding electron-withdrawing ligands, but otherwise changing ligands makes little difference as long as the electronic structure is similar. This can be seen for different aryl and alkyl apical substituents in [166]. El Ghachtouli et al. reported that the exchange of ligands between fluorine and phenyl or methyl groups has negligible effect on i-V behavior, although the ligands go from strongly electron-withdrawing fluorine, via moderately electron withdrawing phenyl to electron donating methyl groups. The electron affinity of the ligands did, however, affect the reduction potential of Co to surface nanoparticles, which in turn improved the HER [167]. Xile Hu et al. reported a more ambiguous effect of manipulating electron affinities by substituting phenyl- for methyl ligands. In this study, a more positive potential for H$_2$ evolution correlated with a decreased activity for electrocatalysis. Complex red-ox behavior was also reported in this study, such as Co(III) hydride intermediates formed upon reduction in acidic media [168]. Zelinskii et al. utilized perfluorophenyl-ribbed substituents to stabilize Co(I) in an effort to enhance the HER, but although the reduced Co(I) was successfully stabilized, the resulting Co-clathrochelate complex was not electrochemically active in the HER [169].

One of the main challenges for non-noble metal catalysts in aqueous electrolyzer cathodes is their stability in harsh acidic conditions. The Co-clathrochelates show good stability in the reported works, exemplified by a stable overvoltage of 240 mV and a faradaic efficiency of 80%, remaining stable for more than 7 h in pH = 2 and at 1 mA/cm$^2$ and 0.9 V [170].

**Table 3.** Summary of full PEM WE cells with EACs as cathodes.

| Cathode (Loading in mg/cm$^2$) | Membrane | Anode (Loading in mg/cm$^2$) | Temp. (°C) | Performance | Stability | Ref. |
|---|---|---|---|---|---|---|
| MoS$_2$ | Nafion 117 | IrO$_2$ (2) | 80 °C | 0.02 A/cm$^2$@1.9 V | Not reported | [84] |
| 47wt.% MoS$_2$/CB (2.5) | Nafion 117 | IrO$_2$ (2) | 80 °C | 0.3 A/cm$^2$@1.9 V | Increasing current density after 18 h | [84] |
| MoS$_2$/rGO (3) | Nafion 117 | IrO$_2$ (2) | 80 °C | 0.1 A/cm$^2$@1.9 V | Not reported | [84] |
| MoS$_x$/CB (3) | Nafion 115 | Ir black(2) | 80 °C | 0.9 A/cm$^2$@2.0 V | Stable current density over 24 h | [105] |
| Mo$_3$S$_{13}$/CB (3) | Nafion 115 | Ir black(2) | 80 °C | 1.1 A/cm$^2$@2.0 V | Current density decreased by more than 100 mA/cm$^2$ after 24 h | [105] |
| MoS$_2$ nCapsules (2) | Nafion 117 | IrO$_2$ (2) | 80 °C | 0.06 A/cm$^2$@2.0 V | Stable current density for 200 h | [106] |
| MoS$_x$/C-cloth | Nafion 117 | RuO$_2$ (2) | 80 °C | 0.3 A/cm$^2$@2.0 V | Not reported | [107] |
| MoS$_x$/C-paper | Nafion 212 | IrO$_2$ (0.1) | 90 °C | 0.37 A/cm$^2$@1.9 V | Stable current density over 4 h | [108] |
| Pyrite FeS$_2$ | Nafion 115 | IrO$_2$ (2) | 80 °C | 1 A/cm$^2$@2.101 V | Stable for 100 h | [145] |
| Greigite Fe$_3$S$_4$ | Nafion 115 | IrO$_2$ (2) | 80 °C | 1 A/cm$^2$@2.130 V | Stable for 100 h | [145] |
| Pyrrholite Fe$_9$S$_{10}$ | Nafion 115 | IrO$_2$ (2) | 80 °C | 1 A/cm$^2$@2.158 V | Stable for 100 h | [145] |
| 30 wt.% Pd/P-doped C (carbon black) | Nafion 115 | RuO$_2$ (3) | 80 °C | 1 A/cm$^2$@2 V | Stable for 500 h | [171] |
| 30 wt.% Pd/N-doped CNTs | Nafion 115 | RuO$_2$ (3) | 80 °C | 1 A/cm$^2$@2.01 V | Stable for 50 h | [172] |
| 30 wt.% Pd/P-doped Graphene | Nafion 115 | RuO$_2$ (3) | 80 °C | 1 A/cm$^2$@1.95 V | Cell voltage increased to 2.0 V after 2000 h | [173] |
| Activated single-wall carbon nanotubes | Nafion 115 | IrRuO$_x$ | 80 °C | 1 A/cm$^2$@1.64 V | Stable for 90 h | [174] |
| Co NPs/N-doped C | Nafion NRE-212 | IrO$_2$ (0.55) | 80 °C | 1 A/cm$^2$@150 mV $\eta$ from Pt/C | Not tested in the full cell. Stable cathode after 10,000 CV cycles @ 100 mV/s | [175] |
| Boron-capped tris (glyoximato) cobalt complexes on carbon black (Co(dmg)/C) 1 mg/cm$^2$ | Nafion 117 | Ir black (2–2.5) | 90 °C | 1 A/cm$^2$@2.1 V | Not reported | [62] |
| [Co(dmgBF$_2$)$_2$]-Vulcan XC72 2.5 mg/cm$^2$ * | Nafion 117 | IrO$_2$ | 60 °C | 0.5A/cm$^2$@1.7 V | No sign of degradation after 60 h @ 0.2 A/cm$^2$ | [165] |
| [Co(dmg)$_3$(BF)$_2$]BF$_4$-Vulcan XC72 2.5 mg/cm$^2$ * | Nafion 117 | IrO$_2$ | 60 °C | 0.65A/cm$^2$@1.7 V | | [165] |
| Co hexachloroclathrochelates impregnated on Vulcan XC72 5–12 × 10$^{-4}$ mg/cm$^2$ ** | Nafion 117 | Ir black | 80 °C | 1 A/cm$^2$@2.15 V | Not reported | [166] |

* Weight of whole complex. ** Weight of catalyst.

### 4.6. Density Functional Theory (DFT) for HER Catalysts

Density functional theory (DFT) is an essential tool for understanding the mechanisms and active sites of novel catalysts as it enables evaluation of the thermodynamics of the individual steps in HER. Modelling reaction barriers is however computationally demanding, and most studies as such, rather adopt a "$\Delta G$ approach". As HER involves both proton transfer and charge transfer, the activity of a catalyst is intrinsically linked to its crystal and electronic structures. In that respect, the hydrogen bonding strength/adsorption energy ($\Delta G_H$) has been widely used as descriptor of catalyst activity. Following the Sabatier principle, too strong or weak interactions with the catalyst surface tends to lower the overall catalyst activity yielding the typical volcano type behavior (Figure 14).

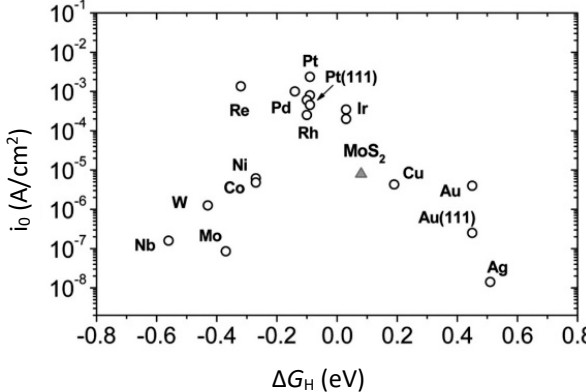

**Figure 14.** Volcano plot of the exchange current density as a function of the DFT-calculated Gibbs free energy of adsorbed atomic hydrogen for nanoparticulate $MoS_2$ and the pure metals. Reprinted with permission from [42]. Copyright 2007 Science.

$MoS_2$ and similar layered transition metal dichalcogenides (TMD) crystallize in two structures, the 2H and 1T polymorphs (Figure 15), with trigonal prismatic and octahedral coordination, respectively.

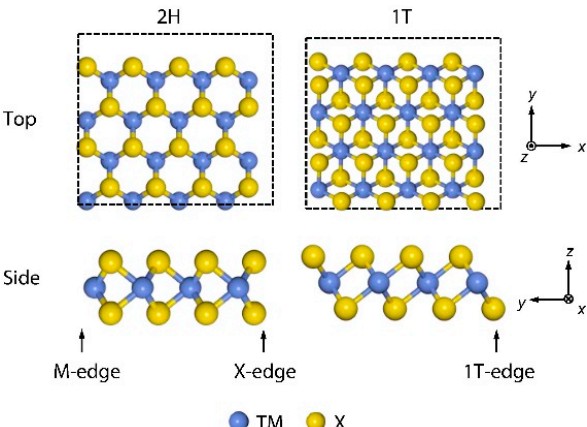

**Figure 15.** Structure of 2H and 1T $MX_2$ dichalcogenides (top view). Reprinted with permission from [176]. Copyright 2015 Elsevier.

The thermodynamically stable 2H polymorph of single layer $MoS_2$ is semiconducting with a band gap of 1.74 eV [177], and its (0001) basal plane exhibits negligible catalytic activity towards HER due to a $\Delta G_H$ of ~2 eV [176]. On the other hand, Hinnemann et al. [91] showed that the ($10\bar{1}0$) Mo edge sites of single trilayer $MoS_2$ can be highly active towards HER, and that they resemble the active sites of the hydrogen-evolving enzymes nitrogenase and hydrogenase [42,91]. The Mo edge exhibits a calculated $\Delta G_H$ of merely 0.08 eV (Figure 16), compared to 0.18 eV of the (10-10) S edge [178], and is as such, close to thermoneutral (for low H coverages). The increased activity of the edge sites

is attributed to in-gap surface states near the Fermi level, implying that 2D $MoS_2$ with a high edge site concentration can be activated towards HER [179]. Significant computational studies have been devoted to exploring strategies to increase the density of activity sites in $MoS_2$, and to optimize $\Delta G_H$ through electronic structure manipulation. Bonde et al. [178] for instance, showed that Co-promotion decreases the $\Delta G_H$ of the S edge to 0.07 eV, but not of the Mo edge, and as such leads to increased number of active sites. Tsai et al. [180] showed that various supports can also be used to tailor the hydrogen bonding to $MoS_2$; for Mo edges. Increasing the catalyst adhesion to the support was found to weaken the hydrogen bonding, and is attributed to downward shifts of the S p-states, which in turn lead to filling of H 1s antibonding states. Efforts have also been made to understand how the basal plane of $MoS_2$ can be activated towards HER through defect chemical, structural and strain engineering [181–184]. Li et al. [181] showed that $\Delta G_H$ of basal plane $MoS_2$ decreases with increasing S vacancy concentration (Figure 16), and that vacancy formation induces in-gap defect states stemming from undercoordinated Mo (Figure 16b), which allows for favourable hydrogen binding. Straining the vacancies was furthermore shown to decrease the $\Delta G_H$ (Figure 16c) even further. Ouyang et al. [184] showed that other native point defects such as $V_{MoS3}$ and $MoS_2$ and extended defects, such as grain boundaries affect hydrogen bonding and as such the HER performance. In addition, Deng et al. [185] showed that single atom transition metal substitution creates in-gap states that lower the $\Delta G_H$, with Pt-$MoS_2$ yielding a close to thermoneutral binding energy.

While the basal plane of 2H-$MoS_2$ is semiconducting [177], its metastable 1T phase is metallic [186] and even its basal plane is highly active towards HER. The metallicity and high HER activity stems from the partially filled Mo 4 d and S states at the Fermi level, leading to favorable $\Delta G_H$ [187]. DFT calculations reveal that $\Delta G_H$ is highly coverage-dependent due to H induced surface reconstructions, reaching values between $-0.28$ and $0.13$ eV for 12.5 to 25 % coverage [187]. The phase stability of the 1T phase, and its band gap and as such HER activity, can be tuned by surface functionalization, by e.g., $-CH_3$, $CH_3$, $OCH_3$, and $NH_2$, which all were shown to bind more strongly to the 1T surface compared to the 2H basal plane [188].

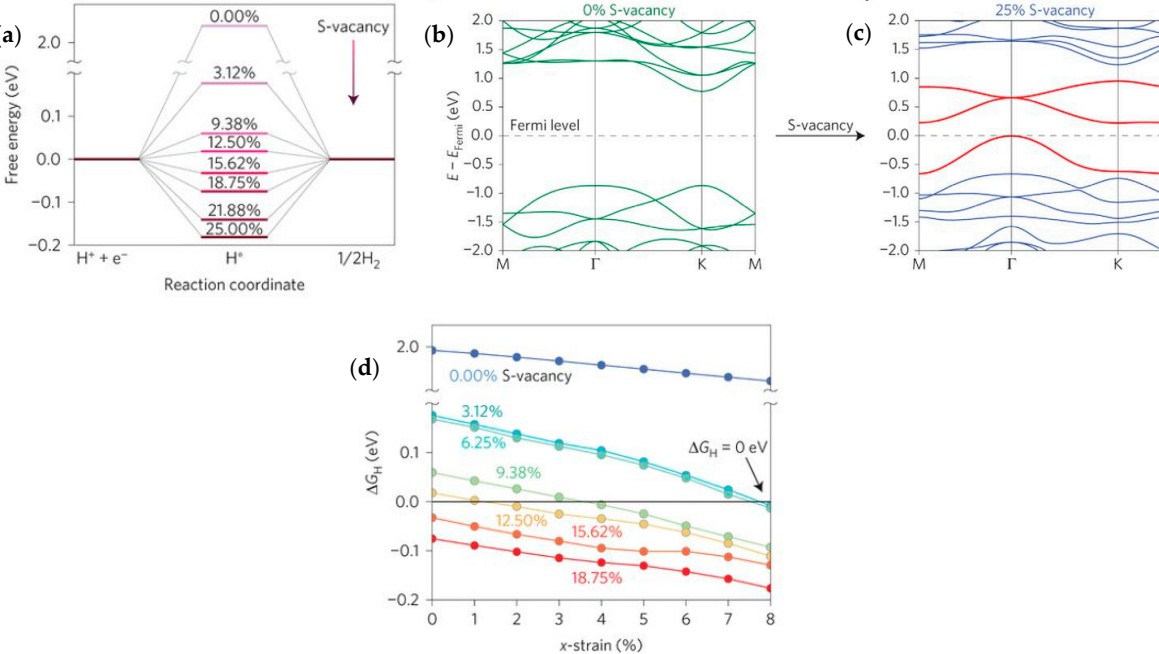

**Figure 16.** (**a**) Free energy vs. reaction coordinate for HER on basal plane $MoS_2$ for various vacancy concentrations. (**b**,**c**) Corresponding band structure, and (**d**) effect of strain and vacancies on $\Delta G_H$. Reprinted with permission from [181]. Copyright 2015 Nature Publishing Group.

Realizing the importance of crystal and electronic structure with respect to HER activity of $MoS_2$, a range of other layered TMD have attracted attention both experimentally and computationally. Tsai et al. [189] showed also that for $MoSe_2$ and $WeSe_2$, the Mo and Se edge sites are more active than the basal planes, and that the selenides generally exhibit weaker H binding than their sulphur counterparts. Tsai et al. [176] furthermore explored the electronic structure, $\Delta G_H$ and the energy of HX adsorption, $\Delta G_{HX}$ (i.e., descriptor for stability) for the basal planes of a range of 2D $MX_2$ (M = Ti, V, Nb, Ta, Mo, W, Pd, and X = S or Se) TMDs. The 2D TMDs vary from semiconducting to metallic (Figure 17), with group 7 TMDs (Mo and W) changing from semiconducting to metallic from the 2H to the 1T phase. The metallic TMD were in general found to exhibit stronger H bonding (lower $\Delta G_H$) than the semiconducting phases (Figure 17). The semiconducting TMDs span a wider range of $\Delta G_H$ than the metallic phases, reflecting the importance of the electronic structure with respect to the HER activity (Figure 17). They found an inverse correlation between $\Delta G_H$ and $\Delta G_{HX}$ for both semiconducting and metallic phases, reflecting the general understanding of the relationship between HER activity and. Furthermore, the metallic TMDs were in general found to exhibit stronger H bonding (lower $\Delta G_H$) than the semiconducting phases.

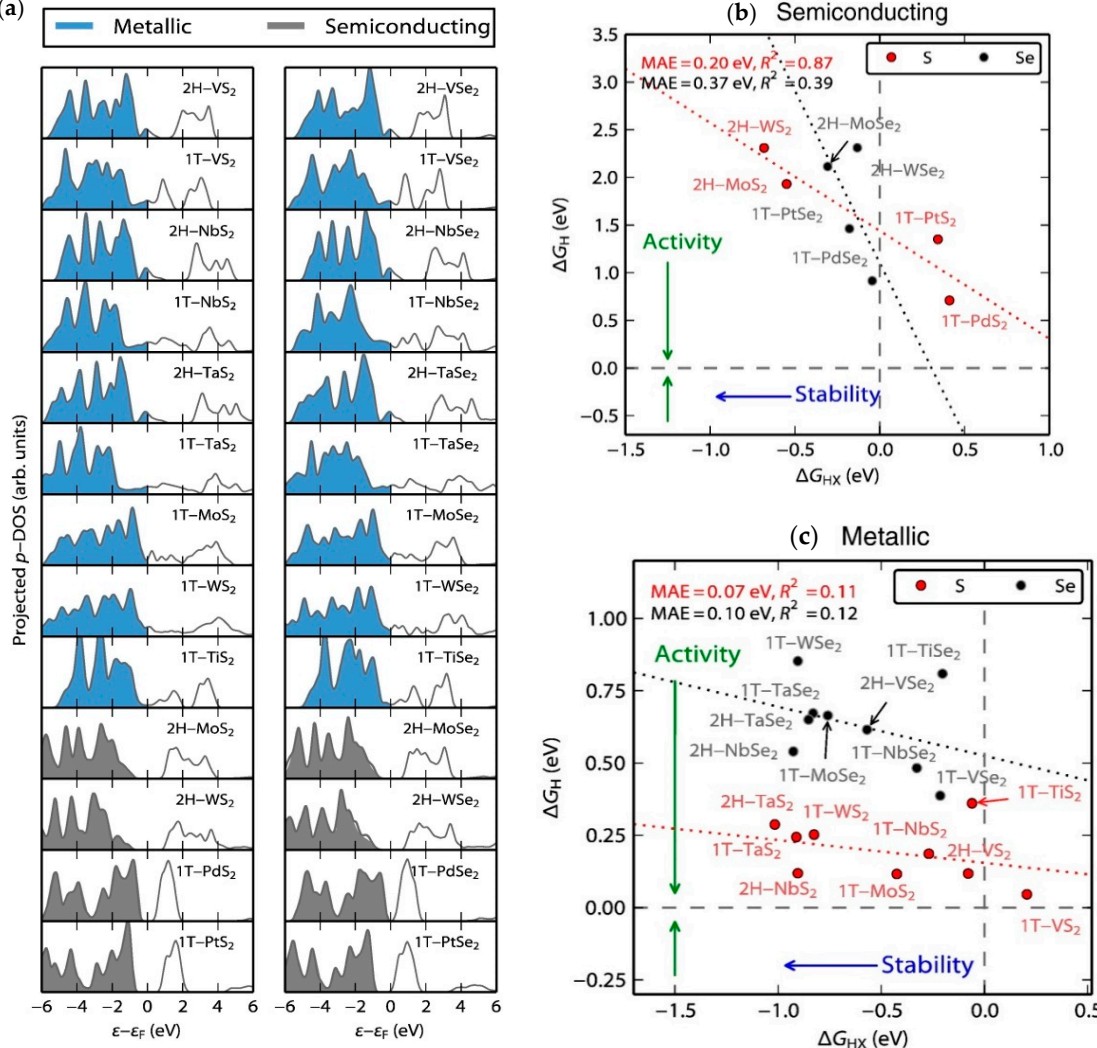

**Figure 17.** (**a**) p-projected density of states on the S or Se atom for 2D TMDs in the 2H and 1T structures relative to the Fermi level, with blue indicating metallic basal planes, while grey ones are semiconducting, (**b**) Semiconducting single-layered TMD basal planes, (**c**) Metallic single-layered TMD basal planes. Reprinted with permission from [176]. Copyright 2015, Elsevier.

Of the HER active transition metal phosphides, those of especially Ni and Mo have been the subject of extensive computational investigations. Bulk $Ni_2P$ is metallic with a crystal structure consisting of alternating $Ni_3P$ and $Ni_3P_2$ planes along the (0001) axis. The HER activity of $Ni_2P$ (0001) surfaces were originally predicted computationally by Liu and Rodriguez [124] showing that the P sites on the phosphide surface play an important role in producing a weak-ligand effect involving Ni $\rightarrow$ P charge transfer, resulting in suitable $\Delta G_H$ and high activity for the dissociation of $H_2$. The bare $Ni_3P$ terminated surface exhibits a strongly binding $Ni_3$ hollow site with $\Delta G_H$ of ~ $-0.5$ eV for the first H, and several sites of lower H binding strength [190,191]. DFT calculations show that the surfaces prefer a P-covered reconstruction of the $Ni_3P$ termination in which a P ad-atom binds on-top of the strongly H binding $Ni_3$ hollow site [190] and that this P ad-atom reduces the bindings strength of the site, and can bind up to 3 H atoms [192,193]. Hakala and Laasonen [191] showed that the H adsorption properties can be modified through Al substitutions, leading to $\Delta G_H$ close to 0 eV.

In a joint experimental-computational effort, Xiao et al. [194] studied hydrogen binding at the Mo, $Mo_3P$ and MoP surfaces showing that 001-Mo surface binds H strongly with a $\Delta G_H$ ranging from $-0.54$ to $-0.46$ eV for $\frac{1}{4}$ to $\frac{3}{4}$ monolayers. The Mo and P terminated (001) MoP surfaces was found to exhibit values of $-0.63$ to $-0.59$ and $-0.36$ to 0.34, respectively, indicating that the P terminated surface can adsorb H at low coverages and desorb at high coverages, reflecting the importance of P also in these catalysts.

## 5. Earth-Abundant Anode Materials

As mentioned previously, the only stable and well-established catalysts for the OER in acidic media are noble metal oxides such as $IrO_x$ and $RuO_x$ [195]. A recent study (2016) on benchmarking of water oxidation catalysts (WOC) revealed that there are no EACs that can reach the target metric of short-term acid stability, which is defined as operation at 10 mA/cm$^2$ for 2 h [196]. We also expected that there are no PEM WE reports based on EACs anodes for the OER side, but this is also not the case. Herein, we report on recent advances and current trends on EACs for the OER that show promising results in terms of performance and stability in acidic media, which exceeded the short-term target of 2 h in just two years. The presented materials and their performance are summarized in Table 4.

Manganese oxide ($MnO_x$) was reported to be functional under acidic conditions and before activation exhibited a Tafel slope of approx. 650 mV/decade, but after potential cycling and activation of the $MnO_x$ film the slope was improved to approx. 90 mV/decade [197]. The authors reported a galvanostatic stability of 8 h in 0.5 M $H_2SO_4$ at a current density of 0.1 mA/cm$^2$ and overpotential of 540 mV. The same group introduced Mn in $CoO_x$ with the former acting as a stabilizing structural element and the $CoMnO_x$ showed a Tafel slope of 70–80 mV/decade and a stability of more than 12 h without any dissolution [198]. The overpotential for a galvanostatic operation at 0.1 mA/cm$^2$, which is 2 orders of magnitude lower than the target values though, was approx. 450 mV. In another work, $MnO_2$ was stabilized by introduction of $TiO_2$ in the undercoordinated surface sites of $MnO_2$. Frydendal et al. applied a 5 nm layer of Ti-modified $MnO_2$ on a 35 nm think layer of pure $MnO_2$ [199]. The composite material exhibited a Tafel slope of 170 mV/decade and a moderate overpotential of approx. 490 mV at 1 mA/cm$^2$. The Mn dissolution in 0.05 M $H_2SO_4$ was suppressed by roughly 50% after the $TiO_2$ modification. The authors came up with this strategy after an initial DFT study, which indicated that guest oxides such as $GeO_2$ and $TiO_2$ should improve the stability of $MnO_2$. The reason is that both $GeO_2$ and $TiO_2$ have lower surface formation energies than $MnO_2$ and are more favorable for termination at the undercoordinated sites on $MnO_2$. Another Mn-containing system is reported by Patel et al., and is based on nanostructured $Cu_{1.5}Mn_{1.5}O_4$:x wt.% F (x = 0, 5, 10, 15) [200]. The $Cu_{1.5}Mn_{1.5}O_4$:10F electrocatalyst in 0.5 M $H_2SO_4$ at 40 °C exhibited an onset potential at 1.43 V vs. RHE for the OER and reached 9.15 mA/cm$^2$ at 1.55 V vs. RHE. Interestingly, the in-house made $IrO_2$ showed the same onset overpotential and 7.74 mA/cm$^2$ at 1.55 V. The reported Tafel slope for the EAC is 60 mV/decade and it should be noted that the current-voltage curves were iR corrected. In a report by Anantharaj et al. it is suggested that the method used to calculate the iR drop compensation

should be reported, along with the uncompensated i-V curves [55]. The material showed also very good stability for almost 24 h of operation at constant current density of 16 mA/cm$^2$. This material is also suitable for the oxygen reduction reaction (ORR), where it showed again similar activity to $IrO_2$. The authors did not apply the $Cu_{1.5}Mn_{1.5}O_4$:10F as a anode in full PEM WE cell, but they did so for the cathode in a PEM fuel cell (PEM FC) mode. The results are promising and the performance is the same as with $IrO_2$ and quite close to the operation in a 3-electrode mode. It should be noted though that the loading of the EAC was 6.7 times higher than for $IrO_2$.

Another Mn-based electrocatalyst was a Mo- and Co-modified electrolytic manganese dioxide (MEMD) developed by Delgado et al. [201]. According to the authors, Mo was incorporated in EMD for improvement of its mechanical stability, while the role of Co was to reduce the water content in EMD and increase the electrical conductivity. The catalyst was synthesized by a simple electrodeposition method and the Mo and Co elements were electrodeposited simultaneously with the EMD. The optimized catalyst had Mo and Co contents of 12.77 wt.% and 0.33 wt.%, respectively, and exhibited an overpotential of 305 mV at 100 mA/cm$^2$ in 2 M $H_2SO_4$. The authors found that this catalyst outperformed a commercial $IrO_2$-based dimensionally stable anode (DSA$^®$) electrode, which under the same conditions had an overpotential of 341 mV. The improved stable performance is explained by the lower charge transfer resistance induced by the Co incorporation, as well as to an increase of the roughness factor. The stability of the catalyst was assessed at a lower overpotential, i.e., at 280 mV, and the degradation of the material was small after approx. 3500 s of operation. In principle, the authors should have reported the stability at 100 mA/cm$^2$ in order to provide more convincing results.

Moreno-Hernandez et al. developed a quaternary oxide, $Ni_{0.5}Mn_{0.5}Sb_{1.7}O_y$, which exhibited an initial OER overpotential of approx. 675 mV vs. RHE in order to reach 10 mA/cm$^2$ in 1.0 M $H_2SO_4$ [202]. The overpotential stabilized at approx. 735 mV and the electrocatalyst performed for 168 h of continuous operation (Figure 18). The authors reported a full cell application in a two-compartment electrolysis cell with Nafion as the separating membrane, but they did not use the catalyst in a full PEM WE cell. The stability of the $Ni_{0.5}Mn_{0.5}Sb_{1.7}O_y$ is comparable to the noble metal oxides and is related to the fact that Ni, Mn and Sb oxides are stable in acidic conditions at OER potentials according to Pourbaix diagrams [203,204].

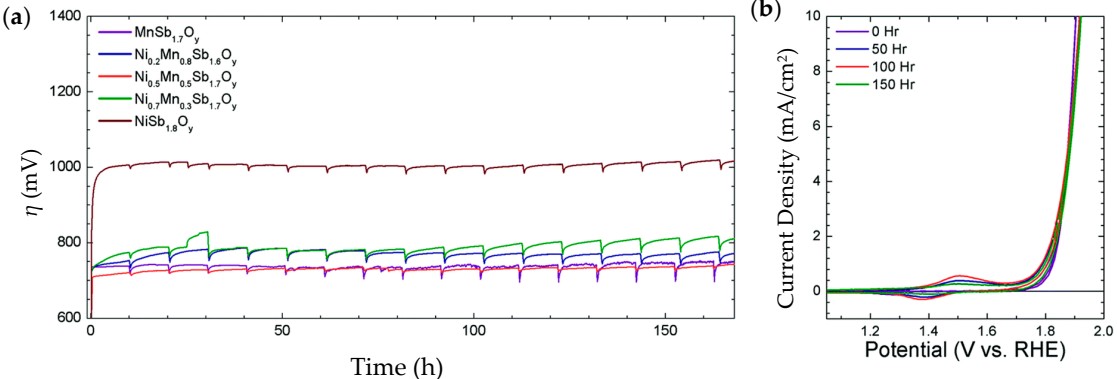

**Figure 18.** (**a**) Stability of the $Ni_{0.5}Mn_{0.5}Sb_{1.7}O_y$ electrodes at 10 mA/cm$^2$ in 1 M $H_2SO_4$. (**b**) Cyclic voltammetry at 10 mV/s in between the stability test. Reprinted with permission from [202]. Copyright 2017 The Royal Society of Chemistry.

Another important element in the aqueous electrochemistry is cobalt (Co). Co oxide-based catalysts have shown excellent performance in alkaline and near neutral pH solution [205,206]. Under strong acidic conditions, however, they show fast dissolution, sluggish kinetics and high overpotentials [196,207,208]. Mondaschein et al. developed a highly crystalline $Co_3O_4$ nanostructured film, which was deposited on FTO by electron-beam evaporation followed by annealing at 400 °C [209]. The overpotential for 10 mA/cm$^2$ was 570 mV vs. RHE in 0.5 M $H_2SO_4$, and the catalyst maintained

an OER with near-quantitative Faradaic yield for over 12 h. Unfortunately, the dissolution rate of Co at this high current density was considerable and further studies are needed for corrosion protection of such structures. To this end, Yan et al. have recently reported the synthesis of mesoporous Ag-doped $Co_3O_4$ nanowires, which showed improved stability over 10 h operation at 1.6 V vs. RHE in 0.5 M $H_2SO_4$, as Ag is known to be stable in acidic media [210]. The Ag-doped $Co_3O_4$ nanowires were synthesized by electrodeposition-hydrothermal process, which was followed by calcination at 400 °C. The nanostructured catalysts showed a Tafel slope of 219 mV/decade and an overpotential of approx. 680 mV at current density of 10 mA/$cm^2$. The authors do not provide any dissolution products analysis or any post-operation analysis of the material, as well as no comparison with $IrO_x$.

Another Co-based catalyst in the form of cobalt-iron Prussian blue-type analogues (PBAs) thin film was developed by Han et al. [211]. In this work, the PB-type water oxidation catalyst promoted the OER over a wide pH range spanning from pH = 1 to pH = 13. According to the authors, an especially interesting aspect of this universal catalyst was the very good behavior at pH = 2. The overpotential required at 10 mA/$cm^2$ was approx. 0.83 V with a Tafel slope of 108 mV/dec. The oxygen evolution was quite close the theoretical one and the catalyst was stable for at least 2 h of operation at 10 mA/$cm^2$. Co-containing polyoxometallates (Co-POMs) have shown promising catalytic properties for water splitting at near-neutral pH [212]. To this end, Blasco-Ahicart et al. developed the Ba salt of Co-phosphotungstate polyanion (Ba[Co-POM], see Figure 19a) that outperformed $IrO_2$ at pH < 1, showing an overpotential of 189 mV vs. RHE at 1 mA/$cm^2$ (Figure 19b) with a faradaic efficiency of 99%. The Tafel slope was 66 mV/decade at the long-term stability was assessed at an overpotential of 250 mV vs. RHE. The initial current was more than 2 mA/$cm^2$ but decreased down to 0.35 mA/$cm^2$ after 24 h of operation. This degradation is assigned to charge localization that reduces the overall performance, which can be retrieved after charge delocalization at open-circuit potential. The authors could not assess the performance of the material at 10 mA/$cm^2$ as the carbon paste, which acted as a binder was not stable.

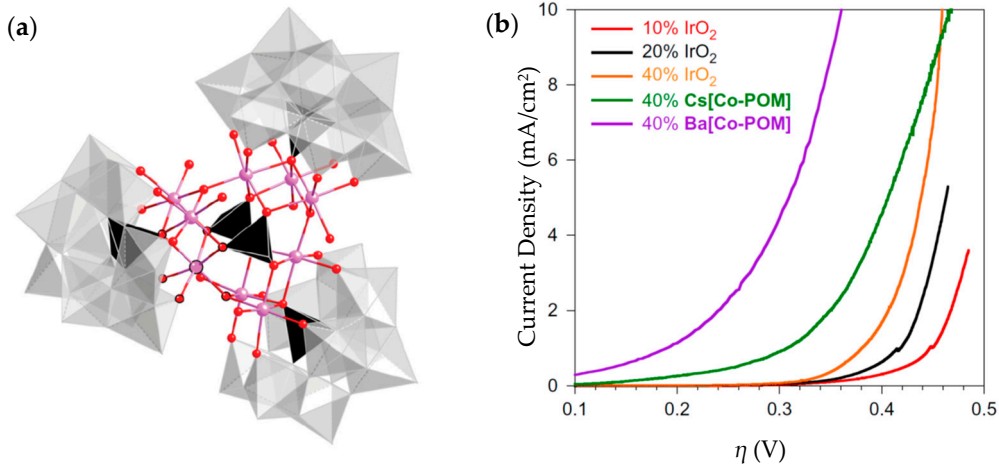

**Figure 19.** (**a**) Molecular structure of the Co-POM cluster. (**b**) Linear sweep voltammetry of different Co-POM electrocatalysts compared with different carbon paste/$IrO_2$ blends in 1 M $H_2SO_4$. Reprinted with permission from [212]. Copyright 2017 Nature Publishing Group, Macmillan Publishers Limited.

An interesting work conducted by Rodriguez-Garcia et al. combines the Co and Sb elements in an anode made of cobalt hexacyanoferrate supported on Sb-doped $SnO_2$ [213]. In this work the synergistic effect of the OER catalysts (CoHFe) and the support, antimonite tin oxide (ATO) is highlighted and the "winning" configuration is when 17 wt.% of CoHFe is deposited on ATO. The onset of the OER was approx. at 1.75 V vs. RHE as determined by RDE experiments. Interestingly, the authors assembled a full PEM WE cell and they have found the onset potential as from the RDE experiments. A current density of the order of 50–100 mA/$cm^2$ was reached at 2 V cell voltage (Figure 20a). The PEM WE cell with this earth-abundant anode showed a rather stable performance at 2 V during a stability testing of

22 h (Figure 20b) The authors studied the Sn and Sb leaching rates during PEM operation and they observed increases leaching rates for cell voltages above 2 V. To our knowledge this the first report on full PEM WE cells using EACs for the anode, showing very promising stability under WE operation.

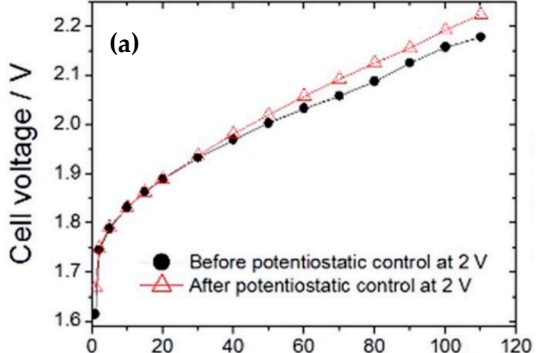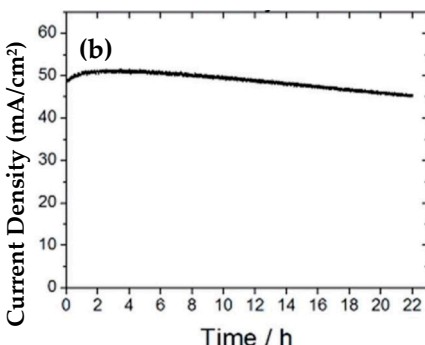

**Figure 20.** (**a**) PEM WE polarization curves before and after 22 h of potentiostatic control at 2 V. (**b**) Stability run at 2 V for 22 h. Reprinted with permission from [213]. Copyright 2018 The Royal Society of Chemistry.

Zhao et al. prepared $FeO_x$ which was incorporated into $TiO_2$ nanowires on Ti foam as the support [214]. The catalyst showed an OER overpotential of 260 mV for 1 mA/cm$^2$ in 0.5 M $H_2SO_4$. The reported Tafel slope was 126.2 mV/decade, while for the $RuO_2$ it was 56.2 mV/decade. The composite material showed very good stability with no significant degradation and after 20 h operation at the OER potential of 1.9 V the current was reduced by 18.7%, but the faradaic efficiency is not provided. Another catalyst involving Fe as the electroactive transition metal is provided by Kwong et al. [215]. In this work, three different Fe-based oxides are studied; the mixed maghemite-hematite, and the single polymorphs, maghemite and hematite. The hematite film was OER-inactive, the maghemite corroded after approx. 6 h of operation, while the mixed polymorph sustained a 10 mA/cm$^2$ for more than 24 h in 0.5 M $H_2SO_4$. The overpotential was 650 mV vs. RHE and increased about 13% after 24 h. The reported Tafel slope is of the order of 56 mV/decade and the faradaic efficiency is almost 100%.

In this paragraph, three more interesting materials for the OER in acid are reported. Yang et al. reported a bifunctional composite material, which is able to catalyze both OER and HER in acidic environment (0.5 M $H_2SO_4$) [216]. A flexible porous membrane comprised of $MoSe_2$ nanosheets on $MoO_2$ nanobelts and carbon nanotubes ($MoSe_2$ NS/$MoO_2$ NB/CNT-M, see Figure 21a) showed a Tafel slope of 112.3 mV/decade and an overpotential of 400 mV at 10 mA/cm$^2$. More importantly, the authors applied the composite porous membrane in a two-electrode water splitting cell and they compared the performance of the EAC against a configuration having $RuO_2$ as the anode and Pt/C as the cathode at a cell voltage of 2 V (Figure 21b). After a large attenuation of the current densities in both configurations the composite porous membrane stabilized at 8.87 mA/cm$^2$ and the noble-metal configuration at 4.38 mA/cm$^2$.

A superaerophobic bifunctional N-doped tungsten carbide nanoarrays catalyst was synthesized on carbon paper with a combination of hydrothermal and CVD methods by Han et al. [217]. The OER onset is at an overpotential of approx. 120 mV vs. RHE, while a high current density of 60 mA/cm$^2$ was reached at approx. 470 mV overpotential. This catalyst outperformed $IrO_2$ in 0.5 M $H_2SO_4$ under 3-electrode configuration as well as in a two-electrode water splitting cell, where both the anode and the cathode were the N-WC nanoarrays (Figure 22c). Unfortunately, the stability of the material is limited and after 1 h of operation at 10 mA/cm$^2$ the overpotential increased from 120 mV to 320 mV vs. RHE, but the faradaic OER efficiency is not reported.

Mondschein et al. reported the intermetallic $Ni_2Ta$ for the OER in 0.5 M $H_2SO_4$ [218]. Intermetallic alloys are metallic conductors and $Ni_2Ta$ has been used as a corrosion resistance coating [219,220].

In their report, Mondschein et al. found that arc-melted $Ni_2Ta$ rods combine the OER activity of Ni and the corrosion resistance of Ta and the intermetallic compound needed 980 mV to reach 10 mA/cm$^2$ (Figure 23b), a behavior assigned to the low electrochemically active surface area (EASA). The authors prepared a polycrystalline Ni-Ta electrode in order to increase the EASA and indeed, the overpotential at 10 mA/cm$^2$ was improved to 570 mV. The polycrystalline electrode showed improved corrosion resistance compared to a Ni pellet electrode prepared in a similar way, as the Ni content in the electrolyte after 36 h operation was below the detection limit of ICP-MS, while for the Ni pellet was 350.5 ppm.

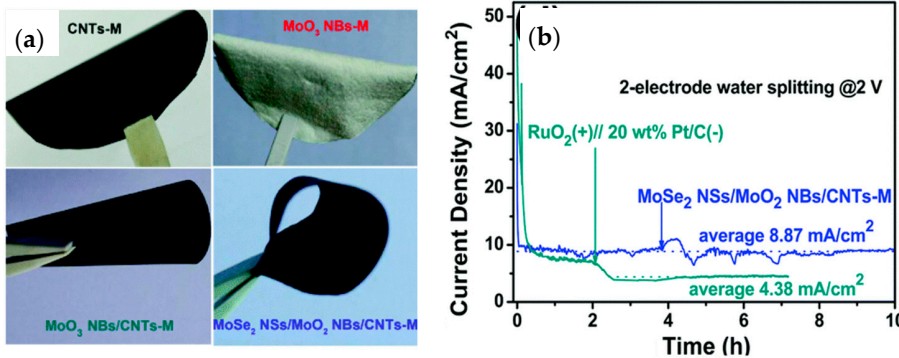

**Figure 21.** (**a**) Photos of the flexible porous membranes of MoSe$_2$ NS/MoO$_2$ NB/CNT-M and the individual components. (**b**) Stability in acidic media using as anode and cathode the MoSe$_2$ NS/MoO$_2$ NB/CNT-M electrode. Reprinted with permission from [216]. Copyright 2018, The Royal Society of Chemistry.

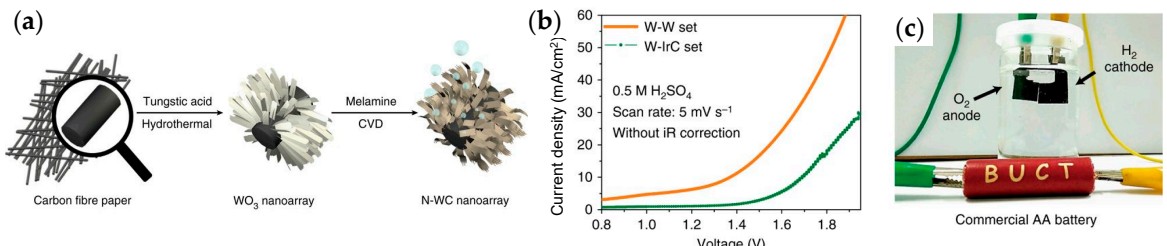

**Figure 22.** (**a**) Synthesis route of the N-doped WC nanoarrays. (**b**) i-V curves of water splitting with the N-WC as anode and cathode electrodes compared with N-WC as the cathode and Ir/C as the anode. (**c**) Video snapshot of the water electrolysis with a 1.5 V commercial battery. Reprinted with permission from [217]. Copyright 2018 Nature Publishing Group.

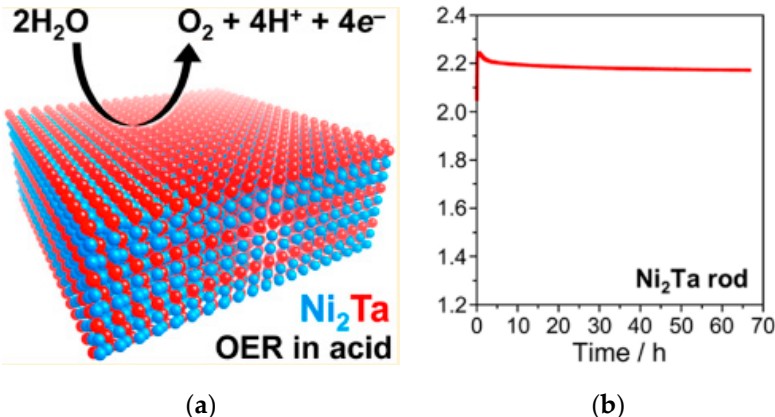

**Figure 23.** (**a**) Intermetallic Ni$_2$Ta for OER in acidic media. (**b**) Galvanostatic measurements of Ni$_2$Ta rods in 0.5 M H$_2$SO$_4$ at 10 mA/cm$^2$. Reprinted with permission from [218]. Copyright 2018 American Chemical Society.



**Table 4.** Summary of the EACs developed for OER in acidic conditions.

| Material | $\eta$ mV | Tafel mV/Decade | Loading | Media | Stability | OER Faradaic Efficiency | Applied in Full PEM WE Cell | Ref. |
|---|---|---|---|---|---|---|---|---|
| Activated $MnO_x$ | 540@0.1 mA/cm$^2$ | 90 | thin film 2–4 nm | 0.5 M $H_2SO_4$, pH = 2.5 | 8 h@0.1 mA/cm$^2$ | ~1% | - | [197] |
| $CoMnO_x$ | 450@0.1 mA/cm$^2$ | 70–80 | films | 0.5 M $H_2SO_4$, pH = 2.5 | 12 h@0.1 mA/cm$^2$ | 91% average | - | [198] |
| Ti-stabilized $MnO_2$ | ~490@1 mA/cm$^2$ | 170 | thin film 40 nm | 0.05 M $H_2SO_4$ | 89%, 1h@1.9V | - | - | [199] |
| $Cu_{1.5}Mn_{1.5}O_4$:10F | 320@9.15 mA/cm$^2$ | 60 | 1 mg/cm$^2$ | 0.5 M $H_2SO_4$ | 24 h@16 mA/cm$^2$ | - | ORR in PEM fuel cell | [200] |
| MEMD | 305@100 mA/cm$^2$ | 115 | roughness factor 429/for DSA 388 | 2 M $H_2SO_4$ | 3500 s@280 mV | - | - | [201] |
| $Ni_{0.5}Mn_{0.5}Sb_{1.7}O_y$ | 675@10 mA/cm$^2$ | 60 | thin film ~300 nm. Ni content 0.48 μmol/cm$^2$ | 1 M $H_2SO_4$ | 168 h@10 mA/cm$^2$. $\eta$ increased to 735 mV | 95% average | - | [202] |
| crystalline $Co_3O_4$ | 570@10 mA/cm$^2$ | 80 | thin film ~300 nm. | 0.5 M $H_2SO_4$, pH = 0.3 | 12 h@10 mA/cm$^2$. Dissolution rate 100 ng/min | above 95% | - | [209] |
| Ag-doped $Co_3O_4$ | 680@10 mA/cm$^2$ | 219 | film, 32.81 m$^2$/g | 0.5 M $H_2SO_4$ | 10 h@6 mA/cm$^2$ | - | - | [210] |
| PB-type | 830@10 mA/cm$^2$ | 108 | film on FTO. | 0.1 M KPi buffer, pH = 2 | 2 h@10 mA/cm$^2$ | close to theoretical@0.3 mA/cm$^2$ | - | [211] |
| Ba[Co-POM] | 189@1 mA/cm$^2$ | 66 | 11 mg | 1 M $H_2SO_4$, pH = 0.2 | From >2 mA/cm$^2$ to 0.35 after 24 h | 99% | - | [212] |
| CoHFe on Sb-doped $SnO_2$ | 780@0.9 mA/cm$^2$ | - | 0.61 mg/cm$^2$ | 0.1 M $H_2SO_4$, | - | - | OER. 50-100 mA@2 V. 6 mA/cm$^2$@1.8 V with 0.5 mg cm-2. Stability: 21 h. Cathode: 0.5 mg/cm$^2$ Pt/C | [213] |
| Fe-$TiO_x$ LNWs/Ti | 260@1 mA/cm$^2$ | 126.2 | 60 mg/cm$^2$ as of $Fe_2O_3$ | 0.5 M $H_2SO_4$ | 20 h@1.9 V. 18.7% current reduction | - | - | [214] |
| Mixed maghemite-hematite | 650@10 mA/cm$^2$ | 56 | 1 mg/cm$^2$ | 0.5 M $H_2SO_4$, pH = 0.3 | >24 h@10 mA/cm$^2$. | ~100% | - | [215] |
| $MoSe_2$ nanosheet/$MnO_2$ nanobelt/CNT (bifunctional) | 400@10 mA/cm$^2$ | 112.3 | 98.46 m$^2$/g | 0.5 M $H_2SO_4$ | 10 h@8.87 mA/cm$^2$ | - | Two-electrode electrolyzer as anode and cathode@2 V | [216] |
| N-doped WC nanoarray (bifunctional) | 470@60 mA/cm$^2$ | - | 10 mg/cm$^2$ | 0.5 M $H_2SO_4$ | 1 h@10 mA/cm$^2$, $\eta$ increases from 120 to 310 mV | - | Two-electrode electrolyzer as anode and cathode@1.4 V | [217] |
| Intermetallic polycrystalline $Ni_2Ta$ | 570@10 mA/cm$^2$ | - | 0.84 cm$^2$ as EASA | 0.5 M $H_2SO_4$ | >66 h@10 mA/cm$^2$. | 85%@20 mA/cm$^2$ | - | [218] |
| Ni42Li205 steel | 445@10 mA/cm$^2$ | 150 | | 0.05 M $H_2SO_4$ pH = 2 | 20 μg/mm$^2$ loss after 14 h@10mA/cm$^2$ | 79%@20 mA/cm$^2$ | - | [221] |

Lastly, Schäfer et al. developed a steel-based catalyst by anodization of Ni42 steel in LiOH [221]. The unmodified Ni42 steel proved to be unstable in acidic media and OER potentials, whereas the LiOH-anodized Ni42 steel was quite stable. The main reason is the oxide formation in the latter, which protects the material from dissolution. This finding was explained by the higher Tafel slope of the treated sample (150 mV/decade) compared to the unmodified steel electrode (88 mV/decade). This supports the concept of the "oxide route" regarding the origin of oxygen formation during electrolysis. A higher activity, i.e., lower Tafel, leads to higher dissolution rates. The optimized sample (Ni42Li205) had an overpotential of 445 mV at 10 mA/cm$^2$ at pH = 0. Moreover, the faradaic efficiency for the OER at 10 mA/cm$^2$ was 79%, and a weight loss of 20 μg/mm$^2$ after 14 h of operation at pH = 1 was observed.

## 6. Summary, Challenges, Perspectives and Future Directions

In this review article, we briefly introduced the energy problem humanity is facing due to the depletion of fossil fuels and the climate changes resulting from their excessive usage. The "hydrogen economy" will become part of our future energy solutions and hydrogen fuel produced by water electrolysis represents a viable, renewable and environmentally friendly option that can replace fossil fuels. We presented a brief technoeconomic analysis and from the learning curves it is estimated that PEM water electrolysis will break even with the cost of hydrogen from fossil fuels around 2030, under an optimistic scenario. Currently, the high cost of hydrogen from PEM WE is related to the polymer exchange membrane, the noble metal electrocatalysts and the high overpotentials for water splitting. With this in mind, we have documented the progress done so far in the discovery and development of EACs both for the OER and HER sides of a PEM electrolyzer. We have not attempted an extensive literature review of EACs, because only for 2017, there were 2043 reports on the development of electrocatalysts. In addition, there are several other reviews, which the reader can refer to in this article, on EACs available in the literature covering either the whole range of new EACs or more specific classes, such as sulfides, phosphides etc. Instead, we reported the state-of-the-art full PEM WE cells based on noble metal catalysts and more importantly, we aimed in documenting how many of the newly developed EACs are actually used in full PEM WE cells, replacing the noble metal-based catalysts. This is equally important during the development stages of any catalyst, in order to observe and record efficiencies, stability and limitations under operating conditions, facts that may differ from the idealized measurements in half cells and rotating disc electrodes (RDE). This point is also emphasized by Spöri et al. in their review for benchmarking OER catalysts [222]. An adequate examination of OER catalysts should be performed both at low and high current densities and moreover, several other parameters should be carefully considered when reporting water oxidation catalysts [55]. For example, iR compensated curves, especially in the high current density region can dramatically differ from the uncompensated ones, reporting overestimated and inaccurate overpotential values. To our surprise, we found only 16 reports on HER EACs employed in full PEM WE cells and only one report for the OER. Of course, the great challenge is to find stable EACs for the OER in acidic environment, as currently the only stable and efficient catalyst is IrO$_2$.

On the other hand, we are among the first to compile the very first EACs with promising efficiencies and stability for the OER under acidic environment. The reader can find the very first 17 breakthrough papers, which we hope that will motivate more research in order to develop and improve the stability of transition metal elements, such as Ni, Co, Fe and Mn for operation under anodic current flow at strongly acidic conditions. Transition metal antimonates of rutile type, as the Ni$_{0.5}$Mn$_{0.5}$Sb$_{1.7}$O$_y$ reported by Moreno-Hernandez et al. show very good stability, which is related to the fact that Mn, Ni and Sb oxides are stable in acid, according to their Pourbaix diagrams [202]. The strategy to integrate unstable catalysts with inactive counterparts, i.e., mixed polymorphs, may lead to stable electrocatalysts. Kwong et al. presented a fine example [215]. The authors combined maghemite and hematite and they achieved a stable operation for more than 24 h at 10 mA/cm$^2$ in 0.5 M H$_2$SO$_4$ at an overpotential of 650 mV vs. RHE, while maghemite and hematite alone are

unstable and not active, respectively. The faradaic efficiency for the OER was also close to 100%. Another strategy is to combine a stable oxide with an unstable one, as the Ti-stabilized $MnO_2$ shown by Frydendal et al. [199]. In this work, a DFT work predicted that $TiO_2$ can be inserted for termination at the undercoordinated sites on $MnO_2$ and in fact, the stability of $MnO_2$ increased by more than 50%. Apart from $TiO_2$, the authors suggested $GeO_2$ as well, as it also has a lower surface formation energy than $MnO_2$.

Intermetallic alloys, such as Ni-Ta, have been used as corrosion protective coatings already from the 90's. Mondschein et al. reported the polycrystalline $Ni_2Ta$ alloy, which was stable for more than 66 h at a current density of 10 $mA/cm^2$ in 0.5 M $H_2SO_4$ [218]. The challenge with such alloys is to increase their surface area by nanostructuring. The same applies for steel-based anodes as presented by Schäfer et al. [221]. The reported Ni42 steel anode modified by a surface oxide layer appears to be both efficient and stable under OER potentials in strong acids. Other steel-based materials should also be tested, especially those containing Fe and Ni, as both elements are active towards OER electrocatalysis. Although surface passivation will increase the charge transfer resistance of the material, on the other hand is a promising trade-off for stable operation.

On the other side, the HER, one can find an enormous amount of EACs both for acidic and basic conditions. We very selectively touched upon the current state-of-the-art and the most promising HER EACs, and our main conclusion is that many more applied systems must be reported. Sixteen works out of thousands make a small sample to draw concrete conclusions from. It is encouraging to see that the HER and OER EACs tested in PEM WE showed similar performances to that expected by measurements in half-cells. There are cases though where the results do not correlate well, as we observed for some Co-clathrochelates. Moreover, in the development of both OER and HER catalysts, it is of utmost importance to report on faradaic efficiencies, as well as post-screening analysis of the electrolyte and catalyst. As we have seen from Tables 3 and 4, it is common to neglect such an analysis. Faradaic efficiency will provide a more accurate and trustable picture of the stability and performance of the catalyst and in addition, will help to identify and distinguish between corrosive and parasitic currents and actual HER and OER currents. There is also a big mismatch on the stability testing times reported on both HER and OER EACs. We generally see that researchers are more confident for the HER catalysts, as the testing times are by far longer than that of OER catalysts. Currently, it is reasonable, as EACs for the OER are in a primitive state with their stability way more challenging. On the other hand, a consensus should be established regarding performance protocols. We encourage the reader to study two benchmarking reviews, which cover a wide range of applied systems for more in-depth information [196,222]. Moreover, a recent perspective article revisits and discusses the validity of ten important parameters that are commonly employed in HER and OER [55].

We take some of the best PEM electrolyzers based on noble metals and EACs, and a valuable comparison is given in Table 5.

**Table 5.** Comparison between full PEM WE cells based on purely noble metal catalysts and those with EACs in the cathode or anode.

| Cathode | Anode | T | Membrane | Current Density | Ref. |
|---|---|---|---|---|---|
| Pt/C 0.1 mg$_{Pt}$/cm$^2$ | Ir$_{0.7}$Ru$_{0.3}$O$_2$ 1.5 mg$_{oxide}$/cm$^2$ | 90 °C | Aquivion ionomer | 1.3 A/cm$^2$@1.6 V | [88] |
| Pt/C 0.5 mg$_{Pt}$/cm$^2$ | Ir$_{0.7}$Ru$_{0.5}$O$_2$ 1.5 mg$_{oxide}$/cm$^2$ | 90 °C | Nafion 115 | 2.6 A/cm$^2$@1.8 V | [73] |
| **Activated single-wall carbon nanotubes (SWNTs)** | IrRuO$_x$ | 80 °C | Nafion 115 | 1 A/cm$^2$@1.64 V | [174] |
| **Mo$_3$S$_{13}$/CB 3 mg$_{Pt}$/cm$^2$** | Ir black(2) | 80 °C | Nafion 115 | 1.1 A/cm$^2$@2.0 V | [105] |
| Pt/C 0.5 mg/cm$^2$ | **CoHFe on Sb-doped SnO$_2$ 3 mg/cm$^2$** | 80 °C | Nafion 115 | 0.05-0.1 A/cm$^2$@2 V | [213] |

It is very encouraging to see that EACs, especially for the HER, have already reached efficiencies very similar to those with noble metal catalysts. Apart from the importance of the transition metals, the non-metallic elements, such as P and S, are also key elements in the development of earth abundant catalysts. DFT works also highlight the noble metal-like activity of the TMD and transition metal phosphides, and in some instances, it is also comparable to the activity and turnover frequencies of enzymes, such as hydrogenases. Furthermore, computational works indicate that P and S, as well as their vacancies, create such an electronic environment that induces favorable binding energies for the adsorption desorption of the H atom.

There is a long way to go for the OER ones, especially concerning their stability, but nevertheless, these results highlight even more the need to employ and operate EACs in full cells. It is also interesting to notice that a PEM WE based on purely EACs can already be realized. It is difficult to say whether the cost of hydrogen from PEM WE breaks even with fossil fuels around 2033, but this review endorses this optimistic scenario. It also provides ways for materials' optimization and development, in order to move forward PEM electrolyzers made purely by EACs, bringing/implying a significant cost reduction to the produced hydrogen.

**Supplementary Materials:** The following are available online at http://www.mdpi.com/2073-4344/8/12/657/s1. Calculations.

**Author Contributions:** X.S. contributed to the principle of operation; K.X. contributed to the documentation of the state-of-the-art PEM WE and EACs for the HER based on carbon materials; C.F. contributed to the documentation of MoS$_2$-based EACs; X.L. to the phosphide-based EACs, M.G. to the FeS$_x$-based ECs, R.S. to the Co-based EACs; T.S.B. contributed to the DFT literature research; T.N. contributed to the writing, editing and original draft preparation and A.C. to the writing, editing and original draft preparation and EACs for the OER.

**Funding:** Financial support from the Research Council of Norway is acknowledged.

**Acknowledgments:** X.S., A.C. and T.N. acknowledge MoZEES, a Norwegian Centre for Environment-friendly Energy Research (FME), co-sponsored by the Research Council of Norway (project number 257653) and 40 partners from research, industry and public sector. K.X. and M.G. acknowledge funding from the Research Council of Norway (RCN) NANO2021 project CO2BioPEC (250261). C.F. acknowledges funding from the Research Council of Norway (RCN) FRINATEK project 2D (262274). X.L. acknowledges funding from the Research Council of Norway (RCN) NANO2021 project EnCaSE (275058). R.S. and T.S.B. acknowledge funding from the Research Council of Norway (272797 "GoPHy MiCO") through the M-ERA.NET Joint Call 2016.

**Conflicts of Interest:** The authors declare no conflicts of interest.

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
