# Peer review of "Earth-Abundant Electrocatalysts in Proton Exchange Membrane Electrolyzers"

_catalysts, doi:10.3390/catal8120657_

Round 1
Reviewer 1 Report
The abstract should be written in a better way. It should be concentrated on novelties.
It is better to improve the resolution and the quality of all figures.
The language in the manuscript can be improved.
It is better to include a detailed stability-comparison between different EACs.
Author Response
Detailed, point-by-point response to the reviewers
Reviewer 1
The abstract should be written in a better way. It should be concentrated on novelties.
The abstract has been fully revised according to the suggestion of the reviewer. It is highlighted in yellow.
It is better to improve the resolution and the quality of all figures.
We have now upgraded the quality of most of the pictures in the article. See for example Figure 19. We have also updated the numbering on each figure in order to appear more homogeneous and with the same font as in the main text. Please notice that some of the figures still appear a bit blurry and this is because the original images are not of good quality either. The figures cannot be highlighted in yellow though.
The language in the manuscript can be improved.
We have gone through the whole document and improved the language, as suggested by the reviewer. As most of the changes were small, we have highlighted in yellow only the most significant ones.
It is better to include a detailed stability-comparison between different EACs.
We agree with the reviewer and a new column has been added in Table 3 regarding the stability of the full cells employing EACs for the HER. We chose to report only on the EACs that have been tested in full cells, as this is the main target of this review article. The stability column was already there for Table 4, which compares the different EACs for the OER. All changes appear in yellow.

Reviewer 2 Report
Non-noble catalysts for water electrolysis is one of the most important challenges for the community, which is already available for the alkaline electrolyte but a long-term missing piece for electrolysis in acid. We believe this review will have an impact in this area.
I recommend it for publication.
But, some questions/concerns should be addressed before publication.
Is it necessary to replace the Pt-Ir based PEM electrolyzer by earth-abundant electrocatalysts? This is one of the most attractive part of this review. But, this reviewer cannot get a clear answer from this manuscript. Of course, the Pt-Ir are noble metals with low abundance. But, the readers will not be satisfied with this description.
Consuming a 100% renewable energy society in Norway based on the “Hydrogen economy”, how much electricity will flow to the hydrogen production by water electrolysis in the future? How many gigawatts or terawatts?
If we choose the Pt-Ir proton exchange membrane electrolyzer, how many years of precious metal annual production would be required?
Please put forward a more detailed description “why earth-abundant electrocatalysts are prerequisite”. The authors could follow the refs “Johnson Matthey. PGM Global Demand Report 2018” and page 14-15 of the SI materials in the paper “Chem. Sci. 2015, 6, 190-196.doi: 10.1039/C4SC02685C”.
In page 5, line 124-125. “the potential needed is higher than the OCV value and will reach typically ~ 1.48 V due to overpotentials related to the OER and HER, as well as to limited ionic conductivity of the electrolyte and system losses.”. Here, the definition of 1.48 V like this is not correct. 1.48 V is the thermoneutral potential.
In “Earth-abundant Anode Material” section. The authors need to add more literature to discuss, for example, a) MnMoCoO in “Sci. Rep. 5, 15208 (2015), doi: 10.1038/srep15208”, b) Electrodeposited electrocatalysts in “J. Am. Chem. Soc. 137, 4347-4357 (2015), doi:10.1021/ja510442p” with a list of earth abundant electrocatalysts in acidic condition in the SI, c) Prussian Blue-type OER catalysts in “J. Am. Chem. Soc. 138, 16037-16045, doi:10.1021/jacs.6b09778 (2016).” Steel-based electrocatalysts in “Catal. Sci. Technol., 2018, 8, 2104-2116” etc.
To identify the real OER catalysts, the stable OER catalysts from the corrosion reaction, the authors should highlight the importance of “Faraday efficiency” and “Detection of the corrosion species” for both the OER and HER catalysts in section “6. Summary, challenges, perspectives and future directions”.
For OER catalysts, faraday efficiency is required for O2 production analysis or both O2 and H2, only H2 analysis is not acceptable, vice versa.
A new test and report protocols should be put forward for the earth-abundant electrocatalysts in acidic condition. Or the old protocols should be highlighted as proposed in “Angew. Chem., Int. Ed., 2017, 56, 5994–6021, doi:10.1002/anie.201608601” and “J. Am. Chem. Soc. 137, 4347-4357 (2015), doi:10.1021/ja510442p”
In page 28 “TiO2=stabilized MnO2” should be “Ti-stabilized MnO2”
Author Response
Detailed, point-by-point response to reviewer.
Non-noble catalysts for water electrolysis is one of the most important challenges for the community, which is already available for the alkaline electrolyte but a long-term missing piece for electrolysis in acid. We believe this review will have an impact in this area.
I recommend it for publication.
But, some questions/concerns should be addressed before publication.
Is it necessary to replace the Pt-Ir based PEM electrolyzer by earth-abundant electrocatalysts? This is one of the most attractive part of this review. But, this reviewer cannot get a clear answer from this manuscript. Of course, the Pt-Ir are noble metals with low abundance. But, the readers will not be satisfied with this description.
We have regarded the Q1, Q3 and Q4 at the same time, as they all consider why we need to replace the PGMs with EACs and how many years of productions are needed with the current state of the art PEM WE systems. We have followed the reviewer’s suggestions and we performed our own calculations as well.
For Q1, Q3 and Q4 we have modified the review as follows and in addition we provide supporting information about our rough calculations. All changes in yellow.
Introduction, page 2: It is established that in terms of efficiency and stability, the platinum group metals (PGMs) are the best choices for electrodes in a PEM electrolyzers, however, the question is at what cost. For example, the annual global production of Pt in 2017 was kg, while the total demand for Pt in the same year was over kg. If the recycled Pt is also considered as part of the production, the annual production of Pt just met the total demand[1]. Therefore, widespread installation of Pt-Ir based PEM electrolyzers will dramatically increase the total demand of PGMs. As an example, a TW hydrogen production system requires 0.5 and 10 years of annual global production of Pt and Ir, respectively [26]. One has also to take into consideration that Ir is typically produced as a minor by-product of Pt [27]. In other words, the annual production of Ir is also determined by the production rate of Pt. As a result, the increasing demands of Ir will increase their cost due to its dependence on Pt mining.
“We performed our own calculations, using the state-of-the-art PEM electrolyzer that we will return to in Chapter 3. In this system, the cathode has 0.4 mgPt/cm2 of Pt, and the anode 1.54 mgIr/cm2 and 0.54 mgRu/cm2 of Ir and Ru, respectively. Our calculations (see SI for more information) suggest that such a PEM system with a power density of 1.18 W/cm2 requires 1.5, 180 and 12 years of annual production of Pt, Ir and Ru, respectively, to cover 1 TW of hydrogen production. It is evident, that the replacement of the noble metal electrocatalysts for both the hydrogen evolution reaction (HER) and oxygen evolution reaction (OER) will have a tremendous impact on the future scale-up activities for PEM WE. Furthermore, competition will be avoided with other industrial activities, such as the automobile and electronics sectors, where the demand for PGMs is big.”
Consuming a 100% renewable energy society in Norway based on the “Hydrogen economy”, how much electricity will flow to the hydrogen production by water electrolysis in the future? How many gigawatts or terawatts?
We are not sure if Norway is interesting for the general readership, however, we found it interesting and we have added this point in the SI.
"The National Transport Plan (NTP) has stated that by 2025, no more fossil cars should be sold. With a yearly sale of roughly 150,000 cars in Norway and assuming that hydrogen cars will have a rising market share from 30% to 50% in the years from 2025 to 2030, the number of hydrogen cars will accumulate to roughly 500,000 by 2030. These cars will then consume around 75,000 tons of hydrogen, which in turn, will require 4 TWh of renewable electric energy."
Source: https://www.openaccessgovernment.org/hydrogen-is-finally-getting-attention-from-norwegian-politicians/47244/
If we choose the Pt-Ir proton exchange membrane electrolyzer, how many years of precious metal annual production would be required?
See Q1 and SI.
Please put forward a more detailed description “why earth-abundant electrocatalysts are prerequisite”. The authors could follow the refs “Johnson Matthey. PGM Global Demand Report 2018” and page 14-15 of the SI materials in the paper “Chem. Sci. 2015, 6, 190-196.doi: 10.1039/C4SC02685C”.
See Q1
In page 5, line 124-125. “the potential needed is higher than the OCV value and will reach typically ~ 1.48 V due to overpotentials related to the OER and HER, as well as to limited ionic conductivity of the electrolyte and system losses.”. Here, the definition of 1.48 V like this is not correct. 1.48 V is the thermoneutral potential.
This point has been corrected in chapter 2.2. All changes in yellow.
“The positive Gibbs free energy change reflects that the water electrolysis reaction is thermodynamically unfavorable. For the reaction to proceed at finite rate, overpotentials for the OER and HER, as well as the electrolyte resistance, must be added to [59]. These represent losses generating heat. At an overpotential of 0.25 V, i.e. an applied cell voltage of 1.48, V this heat balances the heat consumed by the reaction under standard conditions, and the cell operates in thermoneutral mode; 1.48 V is termed the thermoneutral voltage and is reasonable to use when calculating the voltage efficiency of the cell. Thus, the actual operating cell voltage is the sum of all the different overpotentials (Equation 5) [58, 64]. ”
“are the overpotentials related to the anode, cathode, ionic conductivity of the electrolyte membrane, and system losses (resistance in contacts, interconnects, current collectors, wires, etc.), respectively.”
In “Earth-abundant Anode Material” section. The authors need to add more literature to discuss, for example, a) MnMoCoO in “Sci. Rep. 5, 15208 (2015), doi: 10.1038/srep15208”, b) Electrodeposited electrocatalysts in “J. Am. Chem. Soc. 137, 4347-4357 (2015), doi:10.1021/ja510442p” with a list of earth abundant electrocatalysts in acidic condition in the SI, c) Prussian Blue-type OER catalysts in “J. Am. Chem. Soc. 138, 16037-16045, doi:10.1021/jacs.6b09778 (2016).” Steel-based electrocatalysts in “Catal. Sci. Technol., 2018, 8, 2104-2116” etc.
We thank the reviewer for the additional reports on OER in acidic media. They are some very interesting findings and we have included them all in the OER part. The paper from McCrory et al. was already cited in the original version of the manuscript. All changes are highlighted in yellow in chapter 5. Table 4 and conclusions have also been updated.
To identify the real OER catalysts, the stable OER catalysts from the corrosion reaction, the authors should highlight the importance of “Faraday efficiency” and “Detection of the corrosion species” for both the OER and HER catalysts in section “6. Summary, challenges, perspectives and future directions”.
This is a very good point, which is also related to the Q8, brought up by the reviewer and we have added the following section in the chapter 6.
“Moreover, in the development of both OER and HER catalysts, it is of utmost importance to report on faradaic efficiencies, as well as post-screening analysis of the electrolyte and catalyst. As we have seen from Tables 3 and 4, it is common to neglect such an analysis. Faradaic efficiency will provide a more accurate and trustable picture of the stability and performance of the catalyst and in addition, will help to identify and distinguish between corrosive and parasitic currents and actual HER and OER currents. There is also a big mismatch on the stability testing times reported on both HER and OER EACs. We generally see that researchers are more confident for the HER catalysts, as the testing times are by far longer than that of OER catalysts. Currently, it is reasonable, as EACs for the OER are in a primitive state with their stability way more challenging. On the other hand, a consensus should be established regarding performance protocols. We encourage the reader to study two benchmarking reviews, which cover a wide range of applied systems for more in-depth information [200, 226]. Moreover, a recent perspective article revisits and discusses the validity of ten important parameters that are commonly employed in HER and OER [55]”
For OER catalysts, faraday efficiency is required for O2 production analysis or both O2 and H2, only H2 analysis is not acceptable, vice versa.
Please, see Q7. This point is included there as well.
A new test and report protocols should be put forward for the earth-abundant electrocatalysts in acidic condition. Or the old protocols should be highlighted as proposed in “Angew. Chem., Int. Ed., 2017, 56, 5994–6021, doi:10.1002/anie.201608601” and “J. Am. Chem. Soc. 137, 4347-4357 (2015), doi:10.1021/ja510442p”
We have connected this point with Q7 and Q8 and both older and newer protocols are cited and discussed. We have added in the document the following:
Page 33 “This point is also emphasized by Spöri et al. in their review for benchmarking OER catalysts [226]. An adequate examination of OER catalysts should be performed both at low and high current densities and moreover, several other parameters should be carefully considered when reporting water oxidation catalysts [55]. For example, iR compensated curves, especially in the high current density region can dramatically differ from the uncompensated ones, reporting overestimated and inaccurate overpotential values.”
Page 34 “We encourage the reader to study two benchmarking reviews, which cover a wide range of applied systems for more in-depth information [200, 206]. Moreover, a recent perspective article revisits and discusses the validity of ten important parameters that are commonly employed in HER and OER [55]”
In page 28 “TiO2=stabilized MnO2” should be “Ti-stabilized MnO2”
This point has been corrected in the manuscript and highlighted in yellow.
We thank the editor and both reviewers for their valuable inputs, contributing to a significantly improved manuscript worthy of publication.
